# Self-condensation-assisted chemical vapour deposition growth of atomically two-dimensional MOF single-crystals

Lingxin Luo[1,2,4], Lingxiang Hou[1,4], Xueping Cui ®[1] ✉, Pengxin Zhan[1,2], Ping He[1,2], Chuying Dai[1,2], Ruian Li[1], Jichen Dong[1], Ye Zou ®[1], Guoming Liu ®[1], Yanpeng Liu ®[3] & Jian Zheng ®[1] ✉

Two-dimensional metal-organic frameworks (MOFs) have a wide variety of applications in molecular separation and other emerging technologies, including atomically thin electronics. However, due to the inherent fragility and strong interlayer interactions, high-quality MOF crystals of atomic thickness, especially isolated MOF crystal monolayers, have not been easy to prepare. Here, we report the self-condensation-assisted chemical vapour deposition growth of atomically thin MOF single-crystals, yielding monolayer single-crystals of poly[Fe(benzimidazole)$_2$] up to 62 μm in grain sizes. By using transmission electron microscopy and high-resolution atomic force microscopy, high crystallinity and atomic-scale single-crystal structure are verified in the atomically MOF flakes. Moreover, integrating such MOFs with MoS$_2$ to construct ultrathin van der Waals heterostructures is achieved by direct growth of atomically MOF single-crystals onto monolayer MoS$_2$, and enables a highly selective ammonia sensing. These demonstrations signify the great potential of the method in facilitating the development of the fabrication and application of atomically thin MOF crystals.

Owing to the quantum confinement in two dimensions, single-layered crystals exhibit many exotic physical and chemical properties that differ from their bulk counterparts, thus triggering an explosion in the search for atomically thin materials[1–8]. Recently, two-dimensional (2D) metal-organic frameworks (MOFs) with inherent molecular pore/cavity structures that consist of metal ions connected by organic ligands have emerged as one increasingly compelling field due to diverse designable structures and tunable properties[9–13]. At the atomic thickness, 2D MOFs feature rapid mass transfer[11], exceptional carrier transport[13,14] and an extremely high proportion of exposed active sites[15,16], and have easily identifiable atomic structures and bonding arrangements for an ideal model to search for precise structure-performance relationship[17]. Because of these unique characteristics, 2D MOFs have shown

great potential as building blocks for significant technologies in applications such as molecular sensing[18], gas separation[11,19], catalysis[15–17] and superconductor[20]. A key towards the fulfilment of these features and notable applications is to reliably prepare atomically thin MOF crystals in high quality. Great effort has thus been devoted to the development of their preparation methods, such as micromechanical exfoliation[10], liquid exfoliation[16,21], and wet-chemical method[14,17]. However, due to the inherent fragility and strong interlayer interactions in the bulk crystals, 2D MOFs obtained by the above methods are usually thick or have a few layers with small size and/or limited crystallinity. It is highly desired to reliably prepare high-quality and large MOF crystals with atomic thickness, especially the isolated monolayers.

[1]Beijing National Laboratory for Molecular Sciences, Key Laboratory of Organic Solids, Institute of Chemistry, Chinese Academy of Sciences, 100190 Beijing, China. [2]University of Chinese Academy of Sciences, 100049 Beijing, China. [3]State Key Laboratory of Mechanics and Control for Aerospace Structures and Institute for Frontier Science, Nanjing University of Aeronautics and Astronautics, 210016 Nanjing, China. [4]These authors contributed equally: Lingxin Luo, Lingxiang Hou. ✉e-mail: cuixueping@iccas.ac.cn; zhengjian@iccas.ac.cn

Chemical vapour deposition (CVD), a typical bottom-up method, is considered to be a robust way to grow 2D materials with controllable thickness, scalable size and high crystal quality. A range of high-quality monolayer 2D materials, such as graphene[22], h-BN[6,23], and transition metal dichalcogenides (TMDs)[24–26], have been prepared by the CVD method. Therefore, it is promising to prepare high-quality monolayer MOFs with large sizes through the CVD technique. Recently, a multistep CVD method has been demonstrated to achieve the growth of MOF thin films (such as ZIF-8)[27], which marks notable progress in the CVD synthesis of MOFs. Nonetheless, several fundamental challenges remain in terms of CVD growth of atomically thin MOF single-crystals. In a typical CVD process for growing inorganic nanomaterials and graphene, atoms can diffuse rapidly over a long distance on the substrate surface at high growth temperatures (≥700 °C), resulting in the generation of large monolayer domains. While in the CVD growth of MOFs, a low growth temperature (typically below 300 °C) and strong interactions between ligand and substrate, as well as the large mass and volume of ligand molecules, together cause the ligand precursor molecules to have a short free path on the substrate surface. Dense nucleation and growth of small grains of tens of nanometres often ensue, resulting in thick polycrystalline films of MOFs. Thus far, the growth of large, single-crystal, atomically thin MOFs, especially for the monolayers, has yet to be realized using the CVD method.

Here, we demonstrate a self-condensation-assisted CVD (SCA-CVD) growth of atomically thin, single-crystal MOF, in which monolayer poly[Fe(benzimidazole)$_2$] single-crystals were obtained with grain sizes up to 62 μm. The self-condensation of the precursor induced by a temperature gradient design during the CVD process is supposed to be critical in mediating the growth of atomically thin, large-sized MOF single crystals. Characterization by transmission electron microscopy (TEM) and high-resolution atomic force microscopy (HRAFM) have revealed the high crystallinity and atomic structures of single-crystals in the monolayer and few-layer MOF flakes. In addition, good compatibility of the SCA-CVD growth of MOF crystals has been shown and enabled atomically thin MOF single-crystals to grow directly onto monolayer MoS$_2$ to create an ultrathin van der Waals (vdW) heterostructure of MOF/MoS$_2$. Integration combining the precise gate effect of MOF crystal and the high sensitivity of monolayer MoS$_2$ is testified in such heterostructure to engender highly selective ammonia sensing.

## Results

### SCA-CVD growth and characterization of atomically thin MOF single-crystals

The poly[Fe(benzimidazole)$_2$], denoted as Fe$_n$(bim)$_{2n}$, is a typical Van der Waals layered MOF. A structural illustration (Fig. 1a, b) of a single-crystal MOF of Fe$_n$(bim)$_{2n}$ shows a layered construction in which distorted tetrahedral Fe(II) centres are linked with bridging bis-monodentate benzimidazole ligands in the $a–b$ plane[10]. We conducted the SCA-CVD growth of the atomically thin layer of Fe$_n$(bim)$_{2n}$ on SiO$_2$/Si substrates using powdered ferrocene and benzimidazole as precursors in a two-zone tube furnace system (see the "Methods" section for details). Monolayer and few-layer Fe$_n$(bim)$_{2n}$ crystals were obtained and first identified under an optical microscope. As shown in Fig. 1c, d and Supplementary Fig. 1, a well-defined rectangle shape was clearly exhibited by the isolated flakes, which is well aligned with the

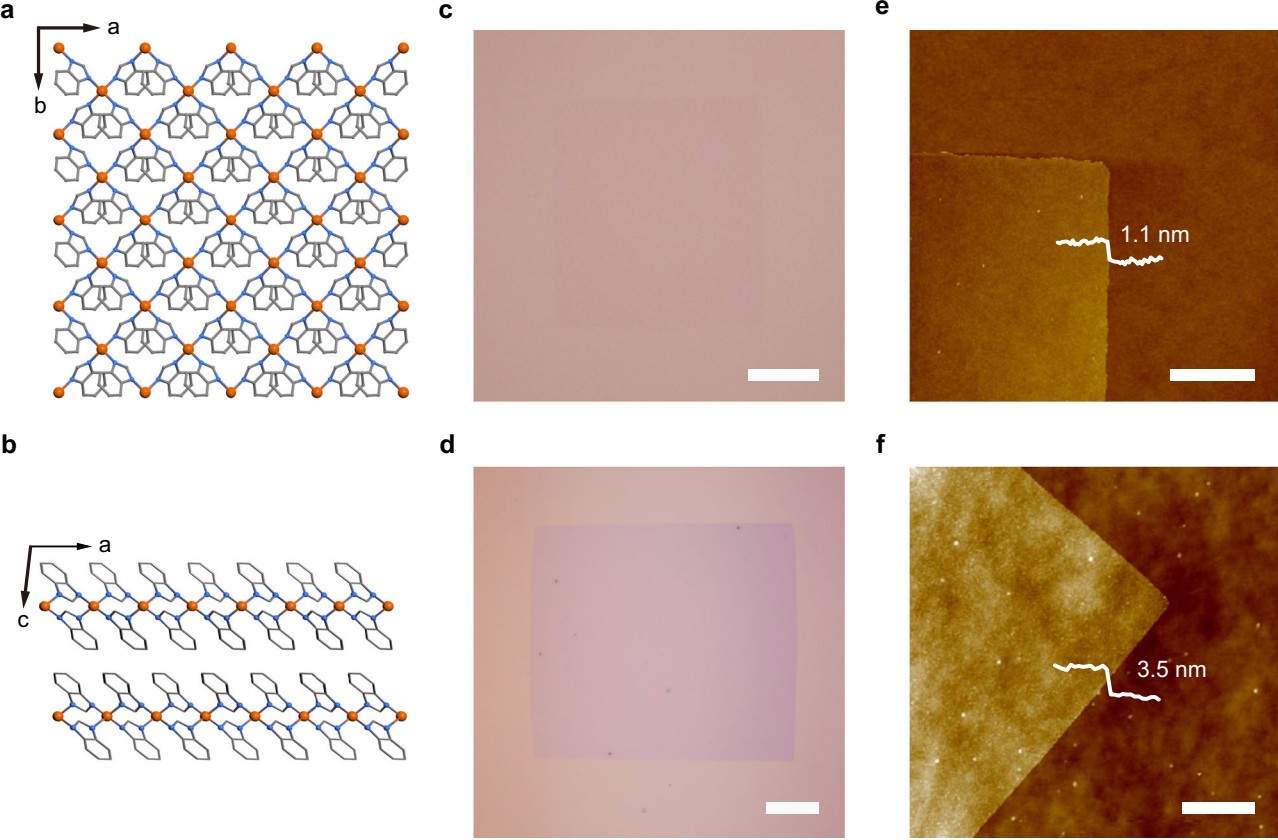

**Fig. 1 | Atomically thin MOF single crystals of Fe$_n$(bim)$_{2n}$. a** Crystal structure of a Fe$_n$(bim)$_{2n}$ single crystal viewed down the $c$-axis ($a–b$ plane), i.e., the lattice structure of a single layer of Fe$_n$(bim)$_{2n}$, where the iron atoms are represented in orange, nitrogen atoms in blue, carbon atoms in grey and hydrogen atoms are omitted for clarity[10]. **b** Layered crystal structure of a Fe$_n$(bim)$_{2n}$ single crystal viewed along the $b$-axis[10]. **c, d** Optical images of a typical monolayer (**c**) and three-layered (**d**) Fe$_n$(bim)$_{2n}$ single crystal grown on a SiO$_2$/Si substrate by SCA-CVD. **e, f** Corresponding AFM images of the Fe$_n$(bim)$_{2n}$ single crystal in (**c**) and (**d**). Scale bars, 10 μm in (**c**), 20 μm in (**d**) and 1 μm in (**e, f**).

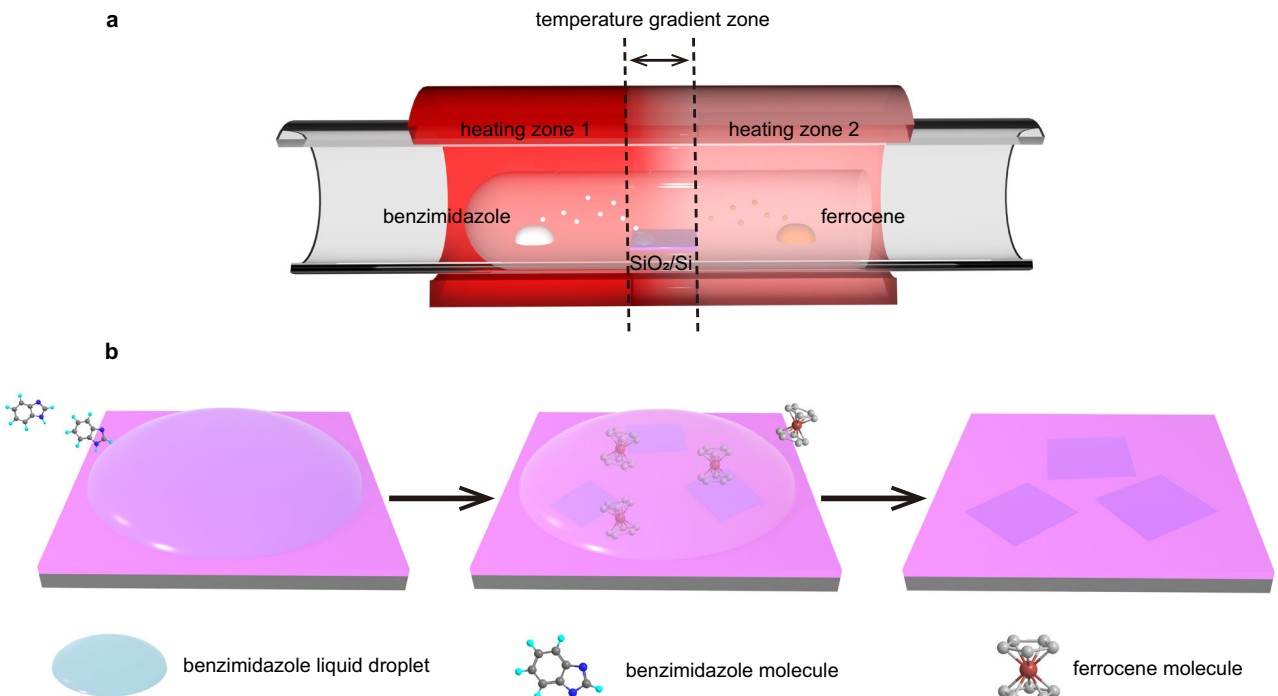

**Fig. 2 | Self-condensation-assisted CVD (SCA-CVD) growth of atomically thin single crystals of $Fe_n(bim)_{2n}$. a** Schematics of the set-up for SCA-CVD growth. **b** The proposed process of SCA-CVD growth involves liquid droplet formation due to the self-condensation of benzimidazole precursor vapour, dissolution and diffusion of the vapourized ferrocene molecules in the liquid droplets, formation of atomically thin single crystals of $Fe_n(bim)_{2n}$.

coordination framework symmetry of $Fe_n(bim)_{2n}$ single-crystals. The exact thicknesses of the flakes were measured to be 1.1 and 3.5 nm (Fig. 1f, g), corresponding to monolayer and three-layer thicknesses, respectively. Statistics based on the AFM measurement showed that the ratio of the monolayer was ~50%. In addition, the $Fe_n(bim)_{2n}$ single-crystals grown by the SCA-CVD method were of considerable lateral dimensions, with lengths for the monolayer and the few-layer up to 62 and 105 μm, respectively. To the best of our knowledge, this was the first time monolayer MOF crystals were obtained with the CVD method, and the MOF crystal was also the largest monolayer single-crystal MOF obtained thus far. Furthermore, the SCA-CVD method allowed atomically thin single-crystal $Fe_n(bim)_{2n}$ to be grown on substrates, including KBr crystals, mica, sapphire and so on, where grown crystals showed a similar appearance to that grown on the $SiO_2/Si$ substrates (Supplementary Fig. 2). Also, atomically thin MOF crystals, including poly[Fe(5-methylbenzimidazole)₂], poly[Fe(5-bromobenzimidazole)₂], poly[Fe(5-chlorobenzimidazole)₂] and poly[Zn(benzimidazole)₂] were successfully prepared via SCA-CVD (Supplementary Figs. 3 and 4).

For SCA-CVD growth of MOFs, ferrocene and various benzimidazole derivatives can be easily obtained in large quantities, facilitating the extension of prepared MOF types, and they are easy to sublimate. The benzimidazole derivatives also demonstrate a suitable melting point in which a temporary liquid environment can be offered. Figure 2 schematically shows SCA-CVD growth. Self-condensation of the precursor into liquid was found to be induced through a temperature gradient design (Fig. 2a) in the CVD process. As shown in Supplementary Fig. 5, obvious vapour-condensed liquid droplets of benzimidazole appeared on the surface of the growth substrate in the gradient zone and on the inner wall of the nearby quartz tube upon thermal sublimation of the precursor. Temperature monitoring (Supplementary Fig. 6) also showed that the substrate temperatures (~200 °C) during the growth stage were higher than the melting point (169–171 °C) of benzimidazole, indicating that the benzimidazole deposited on the substrate existed in liquid state at the growth stage. To further investigate the growth process, a

control experiment was carried out. The self-condensed liquid droplets at different growth times were extracted from the same area of the growth substrate and analysed by gas chromatography–mass spectrometry (GC–MS) (Supplementary Fig. 7). A time-dependent evolution of components in the liquid droplets was revealed, in which the droplets gradually changed from pure benzimidazole into a mixture of benzimidazole and ferrocene. These experimental results depicted a probable process of SCA-CVD growth (Fig. 2b). First, benzimidazole precursor was sublimated into vapour, diffused to the substrate, and then condensed into liquid upon a negative temperature gradient. The vapourized ferrocene subsequently dissolved and diffused in the benzimidazole droplets, accompanied by coordination reactions with benzimidazole molecules. Lastly, the $Fe_n(bim)_{2n}$ nucleated and grew into crystals on the substrate. It should be noted that the involved liquid droplets in the SCA-CVD growth are formed upon vapourization to condensation, and thus they are of high purity and can be fully removed from the growing crystals by a simple vacuum treatment due to the volatility of the precursors without leaving impurities, which is completely different from ordinary solvents and in situ formed liquids.

To examine the crystal quality and structure of the SCA-CVD-grown $Fe_n(bim)_{2n}$ flakes, TEM, selective area electron diffraction (SAED), and HRAFM characterizations were performed. $Fe_n(bim)_{2n}$ flakes grown on KBr crystals were employed for TEM observation, as the excellent water solubility of KBr crystals allows $Fe_n(bim)_{2n}$ flakes to be gently transferred to TEM grids in water, thus preserving intrinsic properties of the transferred flakes. As shown in Fig. 3a, b, monolayer and few-layer $Fe_n(bim)_{2n}$ flakes exhibit typical rectangular shapes with clean and uniform surfaces, which are consistent with the observations obtained by the optical microscope. Collected along the [001] axis, SAED patterns of these flakes (inset of Fig. 3a, b) show clear Bragg diffraction signals and only one set of quasi-fourfold symmetry diffraction spots, which is in good agreement with the simulated electron diffraction pattern (Supplementary Fig. 8), suggesting that both the single-layer and few-layer $Fe_n(bim)_{2n}$ flakes are single-crystals with high

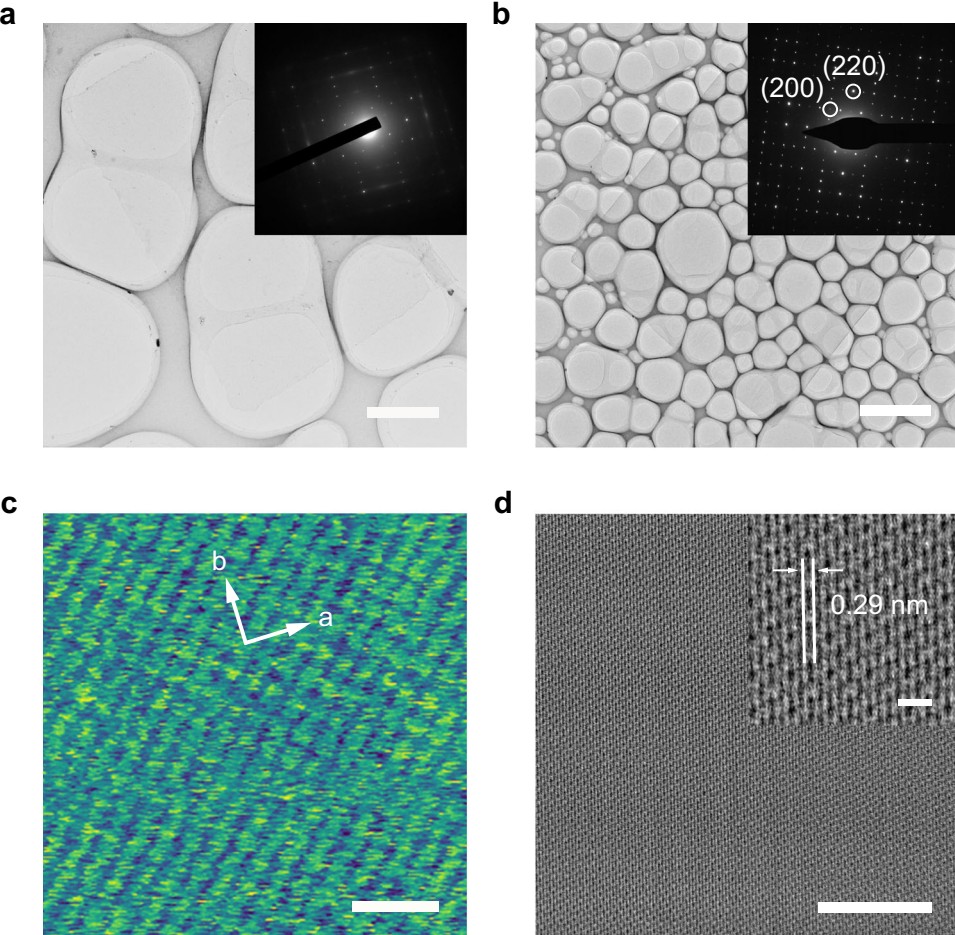

**Fig. 3 | Crystal structure of atomically thin single crystals of $Fe_n(bim)_{2n}$. a** Low-magnification TEM image of a typical monolayer $Fe_n(bim)_{2n}$ single crystal. The inset shows the corresponding SAED pattern in (**a**). **b** Low-magnification TEM image of a typical few-layer $Fe_n(bim)_{2n}$ single crystal, with its SAED pattern shown in the inset. **c** HRAFM image of a typical monolayer $Fe_n(bim)_{2n}$ single crystal. **d** Cryogenic TEM image of a typical few-layer $Fe_n(bim)_{2n}$ single crystal. The inset shows the higher-magnification image, corresponding to the (002) plane of the $Fe_n(bim)_{2n}$ crystal. The cryogenic TEM was conducted under a low dose of 17.56 e/Å²/s. Scale bars, 2 μm in (**a**), 5 μm in (**b**), 2 nm in (**c**), 10 nm in (**d**) and 1 nm in inset of (**d**).

crystallinity. An individual $Fe_n(bim)_{2n}$ flake in a rectangle shape has been further confirmed as a single crystal by collecting SAED patterns at different positions throughout the entire flake (Supplementary Figs. 9–11). A regular periodic lattice structure is displayed in the few-layer $Fe_n(bim)_{2n}$ flake, as shown in the cryogenic TEM image (Fig. 3d). The lattice spacing was measured to be ~2.9 Å, close to the $d_{220}$ of the $Fe_n(bim)_{2n}$ crystal, which corresponds to (002) plane of the $Fe_n(bim)_{2n}$ crystal. Almost no point defects and voids were initially observed within the entire field of view, further confirming high crystallinity of the $Fe_n(bim)_{2n}$ flake. As monolayer MOFs suffer visible and rapid dis-ordering under electron beam irradiation, which is common for 2D MOF crystal characterization with high-resolution TEM, we examined the atomic structure of monolayer $Fe_n(bim)_{2n}$ with HRAFM. As shown in Fig. 3c and Supplementary Fig. 12, rectangular $Fe_n(bim)_{2n}$ units interconnecting into a periodically arranged network with an internal angle of about 90° are clearly observed in the monolayer. The lattice constants of $a$ and $b$ were measured to be 8.3 and 8.4 Å, respectively, which matched well with the crystallographic data, suggesting good quality of the grown monolayer $Fe_n(bim)_{2n}$. The ele-mental information and chemical bonding of $Fe_n(bim)_{2n}$ were also confirmed by scanning transmission electron microscopy (STEM) energy-dispersive X-ray spectroscopy (EDS) mapping images (Supplementary Fig. 13), X-ray photoelectron spectroscopy (XPS, Supplementary Fig. 14) and Fourier transform infra-red (FT-IR, Sup-plementary Fig. 15).

## Mechanism of SCA-CVD growth of single-crystal 2D MOFs

Compared with previous studies, the SCA-CVD method demonstrates the following advantages: it has achieved the growth of large-sized single-layer MOFs and a relatively high single-layer ratio by simple one-step operations, and the grown 2D MOFs single crystals are clean and of good crystallinity. The involvement of the self-condensation liquid in the CVD process is supposed to play a vital role in enabling the growth of the atomically thin large-sized $Fe_n(bim)_{2n}$ single-crystals. During the traditional CVD process for 2D MOF crystals growing, the coordination of precursors in the gas phase is difficult without cata-lysts; the low growth temperature typically leads to irreversible crystal growth, resulting in products with poor crystallinity, and kinetic lim-itations caused by small diffusion rates and radii of the localized ligand precursors also tend to generate small grains. In the SCA-CVD method for MOF growth, a pure temporary liquid environment is introduced into the growth system through the self-condensation of the pre-cursor. This liquid environment serves as a buffer layer to reduce disturbance inside and outside the liquid on the crystal nucleation and growth and allows precursor molecules to be uniformly and stably distributed within the liquid. An increased diffusion rate and radius of the precursor molecules compared with that on the gas–solid interface can be achieved in the liquid to lessen the kinetic limitations of large grain growth along the lateral direction. In addition, similar to most 2D materials, the atoms in a $Fe_n(bim)_{2n}$ layer interact through strong chemical bonds, while the interactions between layers are weak van

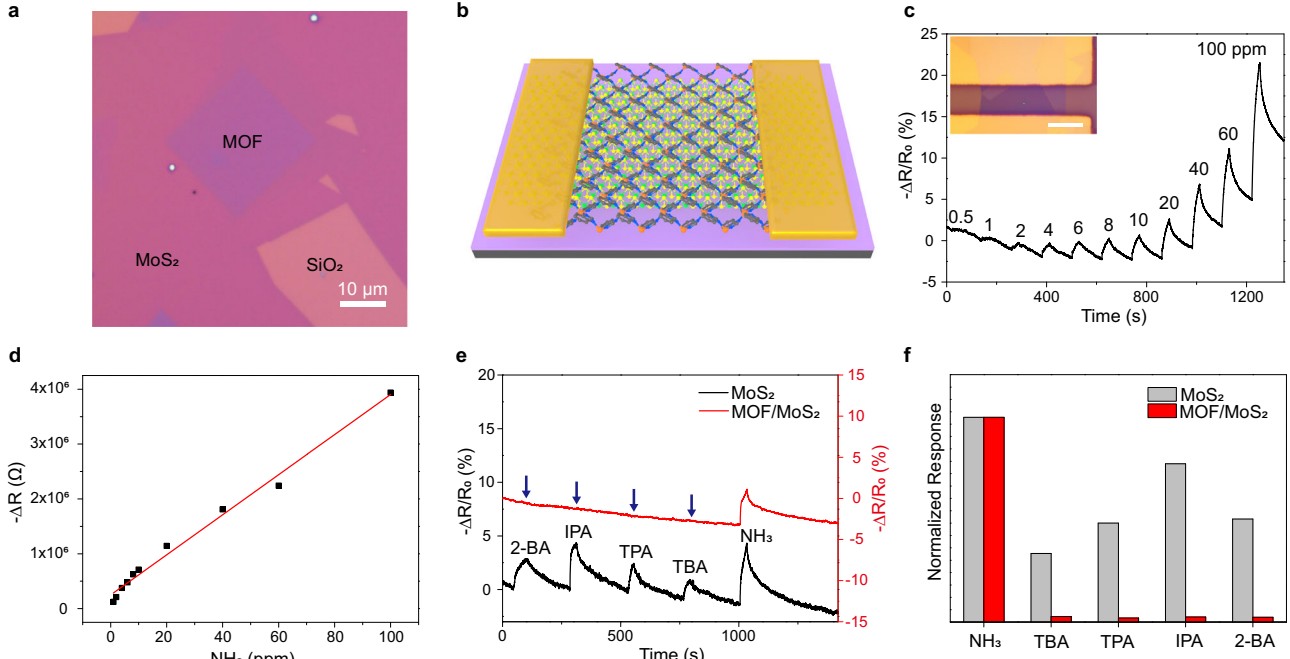

**Fig. 4 | Ultrathin van der Waals (vdW) heterostructure of $Fe_n(bim)_{2n}/MoS_2$.**
**a** Optical image of an ultrathin vdW heterostructure of $Fe_n(bim)_{2n}/MoS_2$.
**b** Schematic illustration of a device based on the vdW heterostructure of
$Fe_n(bim)_{2n}/MoS_2$, in which the periodic network of $Fe_n(bim)_{2n}$ represented by the
dark grey grid is superimposed on a monolayer $MoS_2$ denoted as atomic structure.
**c** Real-time sensing behaviour of a $Fe_n(bim)_{2n}/MoS_2$ sensor upon consequent $NH_3$
exposures at various concentrations. The inset shows an optical image of the
$Fe_n(bim)_{2n}/MoS_2$ sensor. $\Delta R$ is the change in sensor resistance, defined as
$R_{Sensor}-R_0$. Scale bar, 20 μm. **d** Plot of resistance change (black solid squares) of

$Fe_n(bim)_{2n}/MoS_2$ sensor as a function of $NH_3$ concentration. The red line indicates
the fitted line. **e** Sensing response of a monolayer $MoS_2$ (black line) and a
$Fe_n(bim)_{2n}/MoS_2$ (red line) when exposed to $NH_3$ gas, tert-butylamine (TBA)
vapour, tert-pentylamine (TPA) vapour, isopropylamine (IPA) vapour and
2-butylamine (2-BA) vapour of the same concentration (20 ppm). The blue arrow
represents the injection of analytes. **f** Normalized sensing response of the mono-
layer $MoS_2$ and the $Fe_n(bim)_{2n}/MoS_2$ heterostructure in (**e**). **c**–**f** are provided as a
Source Data file.

der Waals force[10]. Previous density functional theory (DFT) studies
have shown that the layer−layer interactions in most van der Waals
materials are around 0.03−0.2 eV per atom, while the strength of
typical chemical bonds is between 2 and 8 eV per atom[28]. The strength
of intralayer interaction is 1−2 orders of magnitude higher than that of
interlayer interaction. Because of the stronger intralayer binding,
attaching atoms to an edge of a $Fe_n(bim)_{2n}$ layer to facilitate the in-
plane extension is much easier than adsorbing atoms on the surface of
a $Fe_n(bim)_{2n}$ layer for the growth of multilayers. More specifically, the
nucleation energy for a new layer is the maximum of $\alpha\sqrt{A} - \beta A$, where
$A$ is the nucleus size, and the first and second terms represent the
positive and negative contributions from the edge and bulk of the
nucleus, respectively[29]. In comparison, the nucleation energy for the
growth of an edge (in-plane extension) is the energy of new dangling
bonds when forming a unitcell of $Fe_n(bim)_{2n}$ on the edge[30]. Under a
near equilibrium growth condition (low $\beta$), the critical nucleation size
for a new layer will be much larger than a unitcell of $Fe_n(bim)_{2n}$, and
more dangling bonds involves in the formation of a new layer, which
results in a higher nucleation energy for a new layer. Therefore,
selective in-plane growth of $Fe_n(bim)_{2n}$ is favourable in an equilibrium
reversible thermodynamic process, as the case in CVD growth of most
2D Van der Waals crystals. Notably, the liquid nature allows dynamic
bonding between the metal and ligand nodes of MOFs, in which the
metal−ligand bonds formed, broke and reformed to correct disorder
or premature structure termination, which is crucial in forming a
crystalline and ordered structure[31]. In addition, as the separate nuclei
in the liquid grow and approach each other, due to the preference of
the free energy of the system to decrease, interface elimination pro-
cesses prefer to occur to reduce the total area of the grain
boundaries[32]. It energetically favoured the formation of large-sized,

single-crystalline MOF crystals in the liquid environment. These ulti-
mately afford the growth of large single-crystal 2D MOFs.

## Selective ammonia sensing of ultrathin vdW heterostructure of $Fe_n(bim)_{2n}/MoS_2$

The demonstrations of high-quality growth of atomically thin
$Fe_n(bim)_{2n}$ flakes make it desirable to explore their integration into
devices. Ultrathin vdW heterostructures constitute a simple but pow-
erful platform for studying fundamental physics and creating func-
tional devices in the 2D limit, in which combining or extending
properties are accessible through the synergy of their constituent
materials[33]. Since the SCA-CVD method is general to a variety of sub-
strates, we fabricated a 2D vdW heterostructure composed of atom-
ically thin $Fe_n(bim)_{2n}$ single-crystal directly grown onto $MoS_2$
monolayers via the SCA-CVD method in which the transfer processes
that often cause quality damage and contamination are not involved.
(Fig. 4 and Supplementary Fig. 16; see the "Methods" section for
details). No obvious interaction between $Fe_n(bim)_{2n}$ and $MoS_2$ was
observed in the heterostructure (Supplementary Fig. 17). Monolayer
$MoS_2$ is characterized by a high sensitivity response due to its high
surface-to-volume ratio and excellent semiconducting properties. To
evaluate such vdW heterostructures, gas-sensing measurements were
performed. The large size of SCA-CVD grown $Fe_n(bim)_{2n}$ single crystals
enables the direct construction of electrodes for the heterostructure
sensors using a hollow mask method without the use of electron beam
lithography technology (see the "Methods" section). Figure 4c shows a
typical sensing response ($S$) of a $Fe_n(bim)_{2n}/MoS_2$ sensor (Supple-
mentary Fig. 18) to varied concentrations of $NH_3$ gas, where $S$ is defined
as $(R_{Sensor}-R_0)/R_0$, $R_0$ and $R_{Sensor}$ are the resistance of the sensor before
and after gas introduction. The sensor responded almost immediately

to $NH_3$ exposure, given that the response is limited by the opening speed of the solenoid valve and the gas mixing speed in the sensing chamber. Clear electrical resistance decreases were observed in the sensors upon $NH_3$ exposure, even at sub-ppm levels (500 ppb). The sensitivity upon $NH_3$ exposure is found to be comparable to that of independent 2D $MoS_2$ sensors[34], which suggests the good sensitivity of the $Fe_n(bim)_{2n}/MoS_2$. In addition, good linear sensor sensitivity can be achieved when the $NH_3$ concentration ranges from 1 to 100 ppm (Fig. 4d), indicating a feasible determination of gas concentration.

Independent monolayer $MoS_2$ is highly sensitive to environmental active stimuli, as their atoms are almost completely exposed; however, it is theoretically impossible to distinguish interfering substances with similar chemical properties by monolayer $MoS_2$. Amine gases/vapours such as $NH_3$, tert-butylamine (TBA), tert-pentylamine (TPA), iso-propylamine (IPA) and 2-butylamine (2-BA) are electron donors; When expose them to a monolayer $MoS_2$, the absorbed molecules on the surface of $MoS_2$ would shift the Fermi level to the conduction band, leading to electrical resistance decreases of $MoS_2$[33]. As shown in Fig. 4e, obvious electrical resistance decreases were all observed in a monolayer $MoS_2$ sensor to exposure of $NH_3$, TBA, TPA, IPA and 2-BA, respectively, with $S$ of −5.45% for $NH_3$, −1.82% for TBA, −2.63% for TPA, −4.21% for IPA and −2.74% for 2-BA (at the same concentration). However, when exposing the above analytes to sensors fabricated with the $Fe_n(bim)_{2n}/MoS_2$ heterostructure, a rapid drop in device resistance only occurs as gaseous $NH_3$ is injected, and almost no change in the resistance is observed with an injection of other amine vapours. The $S$ of the $Fe_n(bim)_{2n}/MoS_2$ sensor for the same concentrations of $NH_3$, TBA, TPA, IPA and 2-BA are about −4.14%, −0.11%, −0.08%, −0.10% and −0.09%, respectively. Calculating selectivity coefficient of the $NH_3$ ($D_{NH_3}$) that is defined as the ratio of responses of $NH_3$ and another interfered gas/vapour, that is $D_{NH_3} = S_{NH_3}/S_i$, there is at least one order of magnitude increase in $D_{NH_3}$ for the $Fe_n(bim)_{2n}/MoS_2$ (Fig. 4f), where $D_{NH_3}$ was 2.99 to TBA, 2.07 to TPA, 1.29 to IPA and 1.99 to 2-BA for the monolayer $MoS_2$ sensor, and 36.95 to TBA, 50.19 to TPA, 40.18 to IPA and 43.32 to 2-BA for the $Fe_n(bim)_{2n}/MoS_2$ sensor. This proves a clearly enhanced $NH_3$ selective response of the $Fe_n(bim)_{2n}/MoS_2$ heterostructure among amine vapours/gases. In addition, the $NH_3$ selectivity of the $Fe_n(bim)_{2n}/MoS_2$ sensor has good reproducibility (Supplementary Fig. 19).

The good sensitivity and selective ammonia response of the ultrathin vdW heterostructure of $Fe_n(bim)_{2n}/MoS_2$ (Supplementary Table 1) reflect the integration of the high sensitivity of monolayer $MoS_2$ and the precise gate effect of MOF crystals. The MOF crystal of $Fe_n(bim)_{2n}$ is known to be a microporous crystalline material with inherent size exclusion characteristics. According to the crystallographic data, the theoretical aperture size of $Fe_n(bim)_{2n}$ unit is estimated to be about 0.21 nm (Supplementary Fig. 20). Considering the structure flexibility of MOF owing to the aromatic rings flip-flop, the effective pore size should be slightly larger than the theoretical aperture size[11,35]. Organic amine vapours, including TBA, TPA, IPA and 2-BA, all have kinetic diameters far exceeding the effective aperture size of $Fe_n(bim)_{2n}$. As expected, due to the size exclusion effect by the small aperture size of $Fe_n(bim)_{2n}$, the gate effect of the $Fe_n(bim)_{2n}$ layer prevents these organic amine vapours from accessing the surface of $MoS_2$, causing almost no gas sensing responses. While $NH_3$ molecules with a kinetic diameter of ~2.6 Å, which is comparable to the effective pore aperture of $Fe_n(bim)_{2n}$, can easily pass through the $Fe_n(bim)_{2n}$ network and reach the $MoS_2$ surface, resulting in gas responses. Notably, when epitaxially grown onto monolayer $MoS_2$ forming a vdW heterostructure, the $Fe_n(bim)_{2n}$ crystals stacked compactly on the $MoS_2$ with an atomically clean interface. In addition, the grown $Fe_n(bim)_{2n}$ crystals were intact single-crystals without grain boundary defects to generate undesired gas permeation pathways. These enable the MOF crystal to maximize its precise gate effect in the heterostructure, which is different from the case of previously reported MOF

thin films. A strict nanochannel created by the $Fe_n(bim)_{2n}$ single-crystals is thus imposed on the monolayer $MoS_2$, in which only molecules of a certain size are able to reach the $MoS_2$. Besides, compared to the long nanopores in a MOF thin film, the $Fe_n(bim)_{2n}$ with atomic-level thickness presents ultra-short nanopores that allow the high sensitivity and fast response of $MoS_2$ monolayers to be retained in the heterostructure, which is responsible for the measured good sensitivity of the $Fe_n(bim)_{2n}/MoS_2$ to $NH_3$ exposure.

In conclusion, our results demonstrate the preparation of high-quality MOF single-crystals with atomic thicknesses using a SCA-CVD method, in which large isolated monolayer single-crystals of $Fe_n(bim)_{2n}$ in high crystallinity were achieved. By directly growing atomically thin MOF single-crystals on monolayer $MoS_2$ via the SCA-CVD method, we fabricated ultrathin vdW heterostructures of $Fe_n(bim)_{2n}/MoS_2$, of which a highly selective ammonia response was shown to result from the synergy of MOF and $MoS_2$. Our methodology inspires a synthetic pathway for creating high-quality atomically thin molecular frameworks that were previously inaccessible and can greatly promote their low-dimensional device integration and applications.

## Methods

### Growth of atomically thin $Fe_n(bim)_{2n}$ single crystals

The single-crystal $Fe_n(bim)_{2n}$ with atomic thickness was prepared by the SCA-CVD method at atmospheric pressure. Precursor powders, benzimidazole (Innochem, 99%, 0.15 mmol) and ferrocene (Adamas Beta, 99%, 0.15 mmol), were respectively loaded at two ends of a single-ended sealed quartz tube. Clean growth substrate, such as $SiO_2/Si$ and sapphire, was placed face up between the benzimidazole and ferrocene. The quartz tube containing precursors and growth substrate was then transferred into a two-zone furnace, with the benzimidazole and ferrocene, respectively, in the centres of the two heating zones and the growth substrate located in the temperature gradient zone. Before growth, the system was first degassed and then purged with ultra-purity nitrogen for 10 min at a flow rate of 200 sccm. The growth recipe was ramped from room temperature to 400 °C within 20 min for the heating zone (1, benzimidazole loading) while from room temperature to 150 °C for the heating zone (2, ferrocene loading) and sit 15 min at 400 and 150 °C. Ultra-purity nitrogen with 5 sccm was flowed in the whole growth process. After growth, the system was degassed to terminate the reaction and then opened the furnace for rapid cooling.

### GC-MS analysis

The liquid droplet components during the SCA-CVD process were analysed by gas chromatography-mass spectrometry GC−MS (SHIMADZU-QP2010) with a packed column DB-5 MS. Droplets located at the same position of the substrates were collected by breaking off the heating programme at desired times. A mixture of benzimidazole and ferrocene with a molar ratio of 1:1 was used as a reference sample for the GC–MS test. All samples for GC−MS testing utilized ethanol as a solvent.

### Preparation of vdW 2D $Fe_n(bim)_{2n}/MoS_2$ heterostructures

We first grew monolayer $MoS_2$ crystals on a $SiO_2/Si$ substrate by CVD using a method reported in ref. 24. The as-grown $MoS_2$ monolayers were transferred to the two-zone furnace as substrates for the growth of $Fe_n(bim)_{2n}$. The SCA-CVD growth of $Fe_n(bim)_{2n}$ was then conducted using the above similar recipes, in which atomically thin $Fe_n(bim)_{2n}$ crystals were directly grown on $MoS_2$ monolayers. The vdW 2D $Fe_n(bim)_{2n}/MoS_2$ heterostructures were thus fabricated.

### Device fabrication and measurements

The gas-sensing devices were fabricated through direct thermal evaporation deposition of 60 nm gold on top of the monolayer $MoS_2$

flakes and $Fe_n(bim)_{2n}/MoS_2$ heterostructures with a metal shadow mask. Gas-sensing experiments were carried out at room temperature in a gas-sensing chamber, where the devices were placed on a chip-carrier with electrically connected gold wires led out by a vacuum terminal block. The gas sensing chamber was maintained at a pressure of 100 Pa and balanced with pure nitrogen at a flow rate of 100 sccm. Analytes we used for the test were $NH_3$ gas, isopropylamine vapour, 2-butylamine vapour, tert-butylamine vapour and tert-pentylamine vapour. Mix them with ultra-pure nitrogen in proportion to achieve desirable concentrations, and then inject them into the gas-sensing chamber by mass flow controllers with a flow rate of 100 sccm. Adjust the flow rate of pure nitrogen to zero when introducing the analyte to maintain a total constant flow rate of 100 sccm. The current of the devices during the gas-sensing test was monitored in real-time at a constant bias voltage (−1.0 V) using the two-probe with an Agilent B2902A Source Meter.

## Characterizations and simulation

Optical images were captured with a Nikon Eclipse LV 100D. AFM images were collected in tapping mode using a Bruker Dimension Icon Scanning Probe Microscope. HRAFM images were recorded on a Cypher VRS1250 in contact mode. The TEM experiments were carried out with a JEOL JEM-2100F operated at 200 kV. Cryogenic TEM image was obtained with a Thermo Scientific Themis 300 with a field-emission gun operating at 300 kV. Simulated SAED pattern was generated with SingleCrystal software.

## Data availability

The data that support the findings of this study are available from the corresponding authors upon request. The source data underlying Fig. 4 and Supplementary Figs. 6, 7, 14, 15, 17 and 19 are provided in the Source Data file. Source data are provided with this paper.

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

## Acknowledgements
We thank B. Guan and J.L. Yue for the assistance with TEM data analysis. We thank Z.P. Wang for the experimental assistance with GC–MS analysis. We thank Z.J. Zhao for the assistance with XPS data analysis. We thank R.J. Feng for the assistance with FT-IR data analysis. We thank M. Wang for the assistance with the grazing incidence wide-angle X-ray scattering (GIWAXS) analysis. The work was supported by the National Natural Science Foundation of China (22175184 (J.Z.), 22105207 (X.C.)), the CAS Project for Young Scientists in Basic Research (YSBR-053 (J.Z.)), the Strategic Priority Research Programme of the Chinese Academy of Sciences (XDB0520202 (J.Z.)) and the CAS Project for Young Scientists in Interdisciplinary Research (J.Z.).

## Author contributions
J.Z. conceived and designed the research. L.L. and L.H. carried out all syntheses and characterization experiments. P.Z., P.H., C.D. and R.L. assisted with the characterization experiments. G.L. assisted with the GIWAXS measurement and data analysis. Y.Z. and Y.L. assisted with the analysis of the characterization results. X.C. and J.Z. fabricated devices and performed the sensing measurement. J.D. conducted the theoretical analysis on the in-plane growth of MOF. J.Z. supervised the research. J.Z., X.C. and L.L. co-wrote the paper. All authors discussed the results and commented on the manuscript.

## Competing interests
The authors declare no competing interests.
