## [Peer Review File · Nature Communications]

Self-condensation-assisted chemical vapour deposition growth of atomically two-dimensional MOF single-crystalsREVIEWER COMMENTS

Reviewer #1 (Remarks to the Author):

In the manuscript entitled "Self-condensation-assisted chemical vapor deposition growth of atomically two-dimensional MOF single-crystals" by Luo et al., the authors developed a new strategy for preparing MOF single-crystals with controlled thickness and explored direct growth MOF-MoS₂ heterostructures for selective ammonia sensing. The strategy developed is of great significance to the MOF community, and the proposed mechanism sounds reasonable. However, the main concerns of this work are (i) the structural characterization of MOF single crystals and (ii) the exemplary application of the resulting MOF-MoS₂ heterostructure. The manuscript should be reconsidered after addressing the following points.

Major points:

- (i) Given that the formation of high crystallinity MOF single crystals is a highlight of this work, more structural characterizations are mandatory, such as grazing incidence wide angle X-ray scattering (GIWAXS), continuous rotating electron diffraction (c-RED), and Fourier transform infrared spectroscopy (FT-IR), to gain a deeper understanding of grain orientation, grain size, precise atomic structure, and chemical bonding.
- (ii) What are the optical and electronic properties of poly[Fe(benzimidazole)₂]? Some basic absorption and PL studies will be very helpful to enhance these aspect.
- (iii) A series of coordination polymers based on Fe(II) centers and different benzimidazole derivatives have been previously reported (Nature Chem 10, 1001–1007 (2018)). Although the strategy presented in this work is different, the structure is not new. In this case, the authors should discuss the advantages of SCA-CVD compared to previous work, and importantly, why they are concerned about poly[Fe(benzimidazole)₂]. A clear motivation seems missing here
- (iv) Although it has been shown that the integration of poly[Fe(benzimidazole)₂] and MoS₂ allows for selective ammonia response, interlayer interaction and electronic coupling in the heterostructure have not been well discussed. More importantly, using monolayer poly[Fe(benzimidazole)₂] single crystals as (passive) coatings to improve gas selectivity does not sound exciting, since many polycrystalline MOF thin films can also be competent for this function. Exemplary applications in the fields of (opto-)electronic, mechanical, and magnetic devices should be considered.

Minor points:

- (i) Given that the preparation is performed in a sealed tube, would it be more reasonable to denote the strategy as "chemical vapor transport method" than "chemical vapor deposition"?
- (ii) Can benzimidazole molecules maintain their original structure in an atmospheric environment of 400 degrees Celsius?

Reviewer #2 (Remarks to the Author):

In this manuscript, a type of atomically thin MOF single-crystal is reported via the self-condensation-assisted chemical vapor deposition (SCA-CVD) method. Impressively, the synthesized monolayer MOF shows a lateral dimension of up to ~40 μm directly imaged by an optical microscope, which will expand the application of ultrathin MOF crystals in device fields. However, I didn't see the flexible and broad generality of the reported synthetic method. The crystallographic characterizations and analyses also shall be enhanced. Consequently, a major revision is recommended for this work and the following suggestions are provided for the authors.

- (1) Is the reported Fe_n(bim)_{2n} a novel MOF structure? If it is, please give the corresponding crystallographic information file (cif). If it is a well-known MOF structure, please provide the related literature.
- (2) For the sake of easy comparisons, the simulated electron diffraction (ED) pattern shall be put together with the experimental SAED.

- (3) Moreover, the elemental mapping results are missing in the TEM characterizations.
- (4) The crystallographic structure of reported $\text{Fe}(\text{bim})_2\text{n}$ is merely verified by SAED. More characterizations must be added in the revised manuscript. For example, low-dose electron microscopy imaging is very useful for MOFs with beam-sensitive properties.
- (5) The generality of the reported synthetic method is suggested to be demonstrated by other types of layered MOF structures.
- (6) Please give a summary and comparison Table to highlight the NH_3 sensing performance obtained by the ultrathin MOF/MoS₂ heterostructure.
- (7) The advantages and disadvantages of the proposed SCA-CVD method shall be compared with other existing ones. If possible, give some demonstrations to highlight the advantages, especially in the field of device applications.

Reviewer #3 (Remarks to the Author):

The manuscript entitled "Self-condensation-assisted chemical vapour deposition growth of atomically two-dimensional MOF single-crystals" proposes a self-condensation-assisted chemical vapor deposition strategy. This strategy enables the growth of atomically thin single-crystal MOF materials (poly[Fe(benzimidazole)₂]) with a size of 42 μm through CVD method for the first time. This strategy has almost no limitations on the substrate and has certain versatility, which provides a reliable basis for subsequent research on MOF material systems. The article also demonstrates the high selectivity of this material in NH_3 sensing, indicating its practical application. However, although the method of this article is innovative, in my opinion, the current data of this article is lacking and it is difficult to fully support its conclusions. Some key issues have not been reasonably explained, and its scientific principles are difficult to understand. I think this article needs a major revision before reconsidering whether it can be published in Nature Communications. Here are some major questions:

1. The most critical problem is that the explanation of the growth of few layers is insufficient. The article explains the principle of reducing the kinetic limitations of lateral growth, but this process does not limit its vertical growth. Therefore, the principle of obtaining few layers or even single layers through this method is still unclear, and I hope the author can reasonably explain the reason of its vertical growth limiting effect.
2. Another question of concern is that the article mentioned that about 50% of the samples are monolayer. Whether a large-area view can be provided to show this result. In addition, can this ratio be regulated by growth conditions? How will the sample change as the growth time is shortened or extended?
3. The article mentioned that the SCA-CVD method has good versatility, but the listed MOF crystals that can be prepared by this method are similar, except for the substituents of benzimidazole. Therefore, whether the method can be described as versatile remains to be reconsidered.
4. In the AFM image of Figure 1f, there are obvious protrusion points. What is its source and whether it will affect the quality of the material.
5. The article mentioned the high selectivity of the material to NH_3 . I wonder how stable the selectivity is this time and whether it can still be maintained after multiple cycles.
6. In addition, the article's explanation for its precise gate effect is that it has no undesired gas permeation pathways. Does this permeation pathway only take effect for ammonia gas? What is the result for small molecules of the gas (such as benzene)?
7. A small issue. The HRAFM results in Figure 3c are not clear enough. FFT processing can be considered.

Reviewer #4 (Remarks to the Author):

1. Would it be possible to use GIWAXS measurements to ensure that the crystal orientation of the MOF ($\text{Fe}(\text{bim})_2\text{n}$) on different substrates are all single crystals?
2. It would be decent to elaborate on more details about the MOF ($\text{Fe}(\text{bim})_2\text{n}$), such as XPS, FTIR, NMR, and TGA. Furthermore, some references to Figures 1a and 1b could be enclosed since insufficient details characterize that MOF.

3. The reference for supporting the properties of $\text{Fe}(\text{bim})_2\text{n}$ would be insufficient since the MOF in that cited article is based on zinc metal. Furthermore, the fabrication process was a top-down process, but not CVD. Therefore, sharing all the physical and chemical properties would fall short of being persuasive. It would be better to analyze pore size distribution and the mechanism of gas selectivity.

4. What is the design of the gas flow orientation inside the experimental apparatus that causes the metal precursor and ligand to flow in opposite directions?

5. Would it be possible to describe how to get the yield of MOF ($\text{Fe}(\text{bim})_2\text{n}$) in as much detail as possible?

6. What would be the criteria for determining whether monolayer single-crystals of $\text{Fe}(\text{bim})_2\text{n}$ up to $42\ \mu\text{m}$ in grain sizes regarding your research? The optical image of Figure 1c cannot be evident to prove that the grain size is $42\ \mu\text{m}$; however, in Figure 3a, the observed crystal size is only $6.5\ \mu\text{m}$ and seems poly-crystal. It could be more convincing with the elaboration.

7. As mentioned in the manuscript, the as-synthesized MOF characterized as single-crystal was grown on the KBr substrate. Nonetheless, the applied device was the one that was grown on MoS_2 . It would be more decent to illustrate that the MOF on MoS_2 was also single-crystal.

RESPONSE TO REVIEWERS' COMMENTS

Reviewer #1 (Remarks to the Author):

Major points:

(i) Given that the formation of high crystallinity MOF single crystals is a highlight of this work, more structural characterizations are mandatory, such as grazing incidence wide angle X-ray scattering (GIWAXS), continuous rotating electron diffraction (c-RED), and Fourier transform infrared spectroscopy (FT-IR), to gain a deeper understanding of grain orientation, grain size, precise atomic structure, and chemical bonding.

ANS: Thank you for your professional advice. We have tried our best to perform GIWAXS characterization on the $\text{Fe}_n(\text{bim})_{2n}$. Two GIWAXS devices, XEUSS and XEUSS 2.0, were used respectively. But no signals have been detected on the $\text{Fe}_n(\text{bim})_{2n}$ flakes (Fig. R1). The atomically thin nature of the $\text{Fe}_n(\text{bim})_{2n}$ is supposed to be responsible for this result. We noted that utilizing GIWAXS technique to characterize atomically thin crystals is extremely rare and has been rarely reported in the field of 2D materials. The GIWAXS measurement is highly likely not suitable for characterizing 2D materials with atomic thickness. To further verify this, GIWAXS measurements were performed on wafer-scale monolayer single-crystal graphene and large-scale monolayer single-crystal MoS_2 using the same equipment. As in the case of $\text{Fe}_n(\text{bim})_{2n}$, no detectable signals were observed in both graphene and MoS_2 (Fig. R1). One of the main reasons for this result may be the energy of the light source used in the GIWAXS measurement is low for atomically thin 2D materials to produce detectable signals. If you insist on this characterization, we can try to schedule a test at the synchrotron radiation center (at Shanghai, China). Its stronger light source possibly leads to detectable signals for the 2D materials. But due to the tight schedule, it may take a long time.

Regarding continuous rotating electron diffraction (c-RED) characterization, we have consulted Professor Junliang Sun from Peking University, who is an expert in the c-RED characterization. The c-RED characterization also requires the preferred thickness of the test material to be 50-500 nm. As thicknesses of the $\text{Fe}_n(\text{bim})_{2n}$ flakes are far below the range, c-RED is failed to obtain the structural information along their thickness direction.

In order to gain a deeper understanding of the grown grain, lots of reference literatures have been looked through. We found that the method of multi-point and multi-region electron diffraction is commonly used to characterize the grain orientation and size of atomically thin materials (Fig. R2, Nat. Mater. 15, 43 (2016); Nat. Commun. 4, 2096 (2013); Nat. Nanotechnol. 15, 289 (2020); Nat. Commun. 6, 8569 (2015); Proc. Natl. Acad. Sci. USA 109, 7992 (2012)). We carefully performed selective area electron diffraction (SAED) on nine different domains of an individual $\text{Fe}_n(\text{bim})_{2n}$ flake. As shown in Supplementary Fig. 9, only one set of quasi-four-fold symmetric diffraction

spots was observed in each domain, indicating that these domains are single crystals. In addition, the lattice orientations extracted from the SAED patterns in the nine domains were found almost identical. This result of single crystals with the same lattice orientation in the domains confirms that the entire $\text{Fe}_n(\text{bim})_{2n}$ flake is a single crystal. Notably, all the individual $\text{Fe}_n(\text{bim})_{2n}$ flakes we have grown are regular rectangles, which matches well with the monoclinic crystal cell structure of $\text{Fe}_n(\text{bim})_{2n}$. Generally, crystals with regular geometric shapes that match their cell structure are considered single crystals, that is, an individual rectangle of $\text{Fe}_n(\text{bim})_{2n}$ flake is a single crystal, as shown by the result of multi-point and multi-region SAED.

Fourier transform infrared spectroscopy (FT-IR) measurements were also conducted to study the chemical bonding information. Supplementary Fig. 14 shows the FT-IR spectra of ferrocene, benzimidazole and $\text{Fe}_n(\text{bim})_{2n}$. It can be clearly seen that the characteristic bands of $3100\text{-}2600\text{ cm}^{-1}$ assigned to N-H vibrations (Inorg. Chim. Acta 23, 155 (1977)) in benzimidazole disappears in the spectrum of $\text{Fe}_n(\text{bim})_{2n}$. It indicates the substitution of N-H in benzimidazole and formation of coordination bonds between Fe and N in $\text{Fe}_n(\text{bim})_{2n}$.

Figure R1. GIWAXS images of $\text{Fe}_n(\text{bim})_{2n}$ nanosheets (a), wafer-scale monolayer single-crystal graphene (b) and large-scale monolayer single-crystal MoS_2 (c).

Figure R2. For 2D materials, conducting SAED measurements at different locations on an individual large-area flake to demonstrate that the flake is a large-sized single crystal is a widely recognized method. Here we listed the relevant literatures. (a) TEM/SAED investigations showing that the graphene prepared by local feeding method is mono-layered and single crystalline. (Nat. Mater. 15, 43 (2016)) (b) Structural characterization of large single crystals of monolayer graphene. (Nat. Commun. 4, 2096 (2013)) (c) TEM characterization of a bilayer graphene film. (Nat. Nanotechnol. 15, 289 (2020)) (d) Structural characterization of monolayer single-crystal WS₂ domains. (Nat. Commun. 6, 8569 (2015)) (e) Raman and TEM characterizations of single-crystalline hexagonal graphene flakes. (Proc. Natl. Acad. Sci. USA 109, 7992 (2012))

Supplementary Figure 9. (a) Low-magnification TEM image of an individual $\text{Fe}_n(\text{bim})_{2n}$ flake. (b-j) SAED patterns in the area enclosed by green circles in a. Only one set of quasi-four-fold symmetric diffraction spots was observed in each area, indicating that these areas are single crystals. The lattice orientations extracted from the SAED patterns in the nine areas were found almost identical. The above results confirm that the entire rectangle flake of $\text{Fe}_n(\text{bim})_{2n}$ is a single crystal.

Supplementary Figure 14. FT-IR spectra of $\text{Fe}_n(\text{bim})_{2n}$, ferrocene and benzimidazole. FT-IR spectroscopy was recorded on VERTEX 70v. The characteristic bands of 3100-2600 cm^{-1} assigned to N-H vibrations² in benzimidazole disappears in the spectrum of $\text{Fe}_n(\text{bim})_{2n}$. It indicates the substitution of N-H in benzimidazole and formation of coordination bonds between Fe and N. (The intensity normalization of infrared spectra was carried out.)

Revisions made:

We have added the description of multi-point and multi-region SAED and FT-IR characterizations in the main text.

“An individual $\text{Fe}_n(\text{bim})_{2n}$ flake in rectangle shape has been further confirmed as a single crystal by collecting SAED patterns at different positions throughout the entire flake (Supplementary Figs. 9 and 10).” (line 153-155)

“In addition, the elemental information... and Fourier transform infrared (FT-IR, Supplementary Fig. 14).” (line 168-172)

Supplementary Figs. 9 and 14 were added in supporting information, with the detailed descriptions in their figure captures.

(ii) What are the optical and electronic properties of poly[Fe(benzimidazole)₂]? Some basic absorption and PL studies will be very helpful to enhance these aspects.

ANS: Thank you for your professional advice. The photoluminescence (PL) and UV-Visible absorption measurements were carried out on the $\text{Fe}_n(\text{bim})_{2n}$, as shown below. No characteristic peaks were observed in the PL spectrum of the $\text{Fe}_n(\text{bim})_{2n}$, except for the peaks for SiO_2/Si substrate. This behavior indicates the $\text{Fe}_n(\text{bim})_{2n}$ is likely an insulator with a large bandgap. To further determine the electronic property, field effect transistors were fabricated on the $\text{Fe}_n(\text{bim})_{2n}$, and found that the electric current of $\text{Fe}_n(\text{bim})_{2n}$ transistors was below the low detection limit of 10^{-12}A . This suggests the $\text{Fe}_n(\text{bim})_{2n}$ is an insulator. The absorption spectroscopy of $\text{Fe}_n(\text{bim})_{2n}$ showed no visible range absorption. It is consistent with the colorless and transparent appearance of $\text{Fe}(\text{bim})_2$ crystals and its structure without a large conjugate system.

Figure R3. (a) PL spectra of $\text{Fe}_n(\text{bim})_{2n}$ grown on SiO_2/Si substrate (black line) and SiO_2/Si substrate (red line) (532 nm excitation, Horiba Labram HR Evolution). (b) UV-Visible absorption spectrum of $\text{Fe}_n(\text{bim})_{2n}$ (Shimadzu UV-2600).

(iii) A series of coordination polymers based on Fe(II) centers and different benzimidazole derivatives have been previously reported (Nature Chem 10, 1001–1007 (2018)). Although the strategy presented in this work is different, the structure is not new. In this case, the authors should discuss the advantages of SCA-CVD compared to previous work, and importantly, why they are concerned about poly[Fe(benzimidazole)₂]. A clear motivation seems missing here.

ANS: Thank you for your advices.

To date, two major strategies have been employed to prepare 2D MOFs: one is top-down method (mechanical or liquid exfoliation from bulk crystals); the other is bottom-up method, such as interfacial synthesis, surfactant-assisted synthesis and CVD. Considerable efforts have been made in preparation of 2D MOFs. However, there are few reports on the preparation of independent monolayer single-crystal MOFs. The mechanical exfoliation method as shown in the work (Nat. Chem. 10, 1001 (2018)) often involves small-size 2D product, low single-layer ratio and repeatability, and a lack of layer number controllability. In addition, due to the fragility of the bulk crystals of MOFs, it is more difficult to exfoliate MOFs crystals into monolayers compared to other 2D materials. In contrast, SCA-CVD method can afford a relatively high single-layer ratio and large-sized single-layer MOFs, contributing to more reliable sample preparation for applications.

The methods of interfacial synthesis and surfactant-assisted synthesis often face difficulties of low crystallinity and surface residual impurities. The SCA-CVD method can provide highly crystalline and clean 2D MOFs single crystals.

Regarding CVD methods for MOFs, big challenges remain. The reported MOFs grown by CVD are most limited to amorphous or polycrystalline films. SCA-CVD has achieved one-step preparation of large-sized monolayer MOF single crystals with simple operations. Moreover, SCA-CVD method is an improvement of the traditional CVD method, which can meet the requirements of preparing MOFs. By modifying reaction conditions, 2D MOF single crystals with different ligands and metal nodes have been prepared by SCA-CVD. In addition, SCA-CVD demonstrates a good substrate generality. 2D MOFs single crystals have been grown on various substrates, even directly on other 2D materials. The directly grown MOFs on 2D materials enabled direct device integration without transfer processes that often cause quality damage and contamination.

We are concerned about Fe_n(bim)_{2n} mainly for the following reasons. The Fe_n(bim)_{2n} is a typical Van der Waals layered MOF. Ferrocene and various benzimidazole derivatives, as the precursors, can be easily obtained in large quantities, facilitating extension of prepared MOF types. In addition, these precursors are very easy to sublime, and the benzimidazole derivatives have a suitable melting point in which a temporary liquid environment can be offered.

Revisions made:

We have added the discussion about the advantages of SCA-CVD as well as the motivation to focus on $\text{Fe}_n(\text{bim})_{2n}$ in the main text.

“Compared with previous studies, SCA-CVD method demonstrates the following advantages: ... of good crystallinity.” (line 173-176)

“in which the transfer processes that often cause quality damage and contamination are not involved.” (line 214-215)

“The large size in SCA-CVD grown $\text{Fe}_n(\text{bim})_{2n}$... of electron beam lithography technology (Methods).” (line 220-223)

“The poly[Fe(benzimidazole)₂], denoted as $\text{Fe}_n(\text{bim})_{2n}$, is a typical Van der Waals layered MOF.” (line 86-87)

“For SCA-CVD growth of MOFs, ... can be offered.” (line 111-115)

(iv) Although it has been shown that the integration of poly[Fe(benzimidazole)₂] and MoS_2 allows for selective ammonia response, interlayer interaction and electronic coupling in the heterostructure have not been well discussed. More importantly, using monolayer poly[Fe(benzimidazole)₂] single crystals as (passive) coatings to improve gas selectivity does not sound exciting, since many polycrystalline MOF thin films can also be competent for this function. Exemplary applications in the fields of (opto-)electronic, mechanical, and magnetic devices should be considered.

ANS: Thank you for your professional advices. The single crystal data analysis revealed that in the heterostructure of $\text{Fe}_n(\text{bim})_{2n}/\text{MoS}_2$, the benzene ring of $\text{Fe}_n(\text{bim})_{2n}$ stands on the surface of MoS_2 , indicating a very weak π electron clouds stacking between $\text{Fe}_n(\text{bim})_{2n}$ and MoS_2 . As interlayer interaction between $\text{Fe}_n(\text{bim})_{2n}$ and MoS_2 also influences the PL of the heterostructure, PL spectra were recorded from the heterostructure and the bare MoS_2 region. As shown in Supplementary Fig. 16, the PL peaks at 1.81 and 1.95 eV respectively matching the A and B direct gap optical transitions of MoS_2 were observed in both heterostructure and MoS_2 . Compared to the bare MoS_2 , no apparent PL shift and quenching occurred in the heterostructure. This suggests no or very weak energy/ charged carrier transfer between $\text{Fe}_n(\text{bim})_{2n}$ and MoS_2 in the heterostructure (Nat. Mater. 13, 1135 (2014); Nat. Rev. Mater. 1, 16042 (2016); Sci. Adv. 2, e1501882 (2016); Nat. Commun. 8, 1906 (2017)). It corresponds to the above single crystal data analysis of the heterostructure. In addition, we measured photoelectric responses of the $\text{Fe}_n(\text{bim})_{2n}/\text{MoS}_2$. Almost no obvious performance change was found in the heterostructure compared with the MoS_2 (Fig. R4).

Magnetic measurement was also performed, as shown in Fig. R5. The Néel temperature of about 20 K was obtained for the $\text{Fe}_n(\text{bim})_{2n}$. A sudden increase in magnetic moment below the Néel temperature was observed, which indicates the occurrence of magnetic transition towards a canted spin structure. These results demonstrate the $\text{Fe}_n(\text{bim})_{2n}$ is

an antiferromagnetic molecule-based magnet caused by spin canting, which agrees with the previous study (Nat. Chem. 10, 1001 (2018)). Generally, magnetic devices are made by ferromagnetic materials, and the antiferromagnetic materials are not suitable for use in magnetic devices.

The Young's modulus of $\text{Fe}_n(\text{bim})_{2n}$ was estimated to be 3–7 GPa, similar to that previously reported for MOFs. (Nat. Chem. 10, 1001 (2018)) This value is much lower than that of graphene (~1000 GPa) and MoS_2 (~270 GPa). Unlike graphene and MoS_2 with strong intralayer bonds, MOFs are composed of intralayer coordination bonds with low bonding density, which can explain the low mechanical strength of $\text{Fe}_n(\text{bim})_{2n}$.

For electronic sensors, their conductivities change when analytes are adsorbed on the active channel, with the magnitude of the change dependent on the electron donating or withdrawing strength of the analyte. Therefore, selective responses in current sensors are mainly realized with analytes with different electrophilic properties, such as NH_3 and NO_2 , NH_3 and H_2 , NH_3 and H_2S , etc. When analytes have similar electron donating or withdrawing strength, conventional electronic sensors are difficult to distinguish them. So far, almost no sensors have been reported to have selective NH_3 response among amine gases/vapours.

We have fabricated the first ultrathin 2D MOF/TMD heterostructures, in which atomically thin single-crystal $\text{Fe}_n(\text{bim})_{2n}$ integrated with monolayer MoS_2 . A highly selective NH_3 response among amine gases/vapours was achieved in this heterostructure, which has not been reported in devices of polycrystalline MOF thin films yet. The polycrystalline MOF thin films share microporous structure and size exclusion characteristic of MOF, but they often suffer from interfacial defects (resulting from interfacial compatibility problems) and internal voids that notably decrease gas selectivity. In contrast, the intrinsic nanoporosity of MOFs is used in the atomically thin single-crystal $\text{Fe}_n(\text{bim})_{2n}$ in which maximizing the size exclusion characteristic for precise molecular and ionic sieving can be achieved.

Our work demonstrates a perfect synergy of 2D TMD material with excellent electrical properties and ultra-thin MOF single crystal with size exclusion characteristic, achieving a sensor with fast and special selective response. This integration and sensor have hardly been reported yet, which is an innovative application.

Supplementary Figure 16. Photoluminescence spectra of $\text{Fe}_n(\text{bim})_{2n}/\text{MoS}_2$ heterostructure (red line) and bare monolayer MoS_2 (blue line). Compared to MoS_2 , no apparent photoluminescence shift and quenching occurred in the heterostructure. This suggests no or very weak energy/charged carrier transfer between $\text{Fe}_n(\text{bim})_{2n}$ and MoS_2 in the heterostructure.

Figure R4. Photoelectric response I/I_0 -time curves of $\text{Fe}_n(\text{bim})_{2n}/\text{MoS}_2$ heterostructure (red line) and bare monolayer MoS_2 (black line) under atmosphere environment and room temperature.

Figure R5. Temperature-dependent magnetization of $\text{Fe}_n(\text{bim})_{2n}$ flakes on quartz. Magnetic measurement was conducted using PPMS DynaCool-9T. The temperature-dependent magnetization of $\text{Fe}_n(\text{bim})_{2n}$ flakes was measured in the range of 2-300 K under an applied magnetic field of 10000 Oe. The Néel temperature of $\text{Fe}_n(\text{bim})_{2n}$ is ~ 20 K. The magnetic moment shows a sudden increase below the Néel temperature, which indicates the occurrence of a magnetic transition towards a canted spin structure.

Revisions made:

Description and discussion of the interlayer interaction in the heterostructure have been added in the main text and supporting information.

“No obvious interaction between $\text{Fe}_n(\text{bim})_{2n}$ and MoS_2 was observed in the heterostructure (Supplementary Fig. 16).” (line 216-217)

Supplementary Fig. 16 was added in supporting information, with the descriptions in figure capture.

Minor points:

(i) Given that the preparation is performed in a sealed tube, would it be more reasonable to denote the strategy as “chemical vapor transport method” than “chemical vapor deposition”?

ANS: Thank you for your question. Allow me to clarify first that the quartz inner tube we used in the SCA-CVD growth is open at one end and sealed at the other end. Such one-end sealed tube is commonly used in CVD process for preparing 2D materials.

(Adv. Mater. 30, 1704674 (2018); J. Am. Chem. Soc. 139, 1073 (2017)) CVD is a method where chemical reaction occurs between precursors and a new chemical substance is thus formed. While in a CVT process, the raw materials are transported from high temperature zone to low temperature zone with the assistance of transport reagents (such as I_2) and large-sized crystals with the same chemical composition as the raw materials are then grown. Besides, CVD is usually an open system, while CVT requires vacuuming and sealing to form a closed system. The SCA-CVD growth involves two types of vapour-phase precursors and in an open system. So, it is a CVD process.

(ii) Can benzimidazole molecules maintain their original structure in an atmospheric environment of 400 degrees Celsius?

ANS: Thank you for your question. Allow me to explain first that the SCA-CVD growth is carried out under an inert atmosphere. We used GC-MS measurement to examine the structure of the treated benzimidazole that was heated at 400 °C for 30 minutes and cooled to room temperature in an inert atmosphere. Fig. R6 displays the GC spectrum and MS spectrum, respectively. Except for the ethanol solvent peak, there is only one signal appeared in the GC spectrum, indicating only one component exists in the test sample. The obtained MS spectrum is almost identical to the standard spectrum of the benzimidazole, confirming the only component in the sample is benzimidazole. These results suggest that benzimidazole molecules can maintain their original structure in the growth atmospheric environment of 400 °C.

If the atmospheric environment in the question means air, the answer is no. As shown in Fig. R7, it can be clearly observed that the color of benzimidazole changed from white to brown after heated at 400 °C in air, indicating oxidation degeneration of the benzimidazole. Therefore, benzimidazole cannot maintain their original structure after heated at 400 °C in air.

Figure R6. The GC-MS spectra of benzimidazole after heated at 400 °C for 30 minutes in an inert atmosphere. Ethanol was utilized as the solvent for GC-MS testing.

Figure R7. The photograph of benzimidazole powders before (a) and after (b) heated at 400 °C in the air environment.

Reviewer #2 (Remarks to the Author):

(1) Is the reported $\text{Fe}_n(\text{bim})_{2n}$ a novel MOF structure? If it is, please give the corresponding crystallographic information file (cif). If it is a well-known MOF structure, please provide the related literature.

ANS: Thank you for your advice. The $\text{Fe}_n(\text{bim})_{2n}$ is not a new MOF structure, and its related literature has been cited as reference 10 (Nat. Chem. 10, 1001 (2018)) in the main text. We have marked this literature again at the structural description section and in the legend of Figure 1.

Revisions made:

We have added the literature in the structural description section in the main text.

“...in the a-b plane¹⁰.” (line 89)

“...omitted for clarity. (ref. 10) **b**, Layered crystal structure of a $\text{Fe}_n(\text{bim})_{2n}$ single crystal viewed along the *b* axis. (ref. 10)” (line 302-303)

(2) For the sake of easy comparisons, the simulated electron diffraction (ED) pattern shall be put together with the experimental SAED.

ANS: Thank you very much for your reminder. Following your suggestion, we have put the simulated electron diffraction (ED) pattern together with the experimental SAED in Supplementary Fig. 8, as shown below.

Supplementary Figure 8. (a) The experimental SAED pattern of a $\text{Fe}_n(\text{bim})_{2n}$ flake. (b) A simulated SAED pattern of $\text{Fe}_n(\text{bim})_{2n}$ flake down the *c* axis.

Revisions made:

We have revised Supplementary Fig. 8 as shown above.

(3) Moreover, the elemental mapping results are missing in the TEM characterizations.

ANS: Thank you for your advice. We have conducted scanning transmission electron microscopy (STEM) energy-dispersive X-ray spectroscopy (EDS) mapping for the $\text{Fe}_n(\text{bim})_{2n}$ flake. As shown in Supplementary Fig. 12, the elements C, N and Fe are distributed uniformly throughout the area of the flake.

Supplementary Figure 12 | STEM-EDS mapping of $\text{Fe}_n(\text{bim})_{2n}$ flake. STEM image of a $\text{Fe}_n(\text{bim})_{2n}$ flake (a) and corresponding elemental maps for C (b), N (c) and Fe (d). STEM-EDS mapping was obtained by a JEOL JEM-F200 with acceleration voltage 200 kV.

Revisions made:

We have added Supplementary Fig. 12 in the main text and supporting information.

“In addition, the elemental information... scanning transmission electron microscopy (STEM) energy-dispersive X-ray spectroscopy (EDS) mapping images (Supplementary Fig. 12)...”
(line 168-172)

(4) The crystallographic structure of reported $\text{Fe}_n(\text{bim})_{2n}$ is merely verified by SAED. More characterizations must be added in the revised manuscript. For example, low-dose electron microscopy imaging is very useful for MOFs with beam-sensitive properties.

ANS: Thank you for your professional advice. Except the characterization of SAED, HRAFM (Fig. 3c) and cryogenic TEM (cryo-TEM, Fig. 3d) were also employed to investigate the crystallographic structure of the atomically thin $\text{Fe}_n(\text{bim})_{2n}$ crystals. The low-dose electron microscopy imaging technology has been already used in the cryo-TEM and a low-dose of $17.56 \text{ e}/\text{\AA}^2/\text{s}$ was applied. Under this condition, HRTEM image was obtained (Fig. 3d), displaying lattice spacing and lattice constants consistent with the crystallographic data of $\text{Fe}_n(\text{bim})_{2n}$.

In order to gain a deeper understanding of $\text{Fe}_n(\text{bim})_{2n}$, XPS and FT-IR characterizations were also performed.

Supplementary Fig. 13 displays the Fe(II) $2p$ core-level and N $1s$ core-level XPS spectra of $\text{Fe}_n(\text{bim})_{2n}$ and precursors. In the Fe $2p$ spectrum of ferrocene, the peaks at 707.8 and 720.6 eV are assigned to the binding energies of the $2p_{3/2}$ and $2p_{1/2}$ orbitals of Fe(II) species, respectively. A weak peak at 711.3 eV can be attributed to a satellite peak, an indicator of Fe(II) in low-spin configuration. The $2p_{3/2}$ and $2p_{1/2}$ orbitals of Fe(II) in the $\text{Fe}_n(\text{bim})_{2n}$ were observed at 710.5 and 723.9 eV respectively, shifting toward higher binding energies than those in the ferrocene. This can be explained by the decrease in electron cloud density around Fe atoms in the $\text{Fe}_n(\text{bim})_{2n}$ compared to ferrocene, as benzimidazole has stronger electron-withdrawing ability than the cyclopentadienyl group of ferrocene. The satellite peaks for $\text{Fe}_n(\text{bim})_{2n}$ appear at 715.0 and 729.2 eV, suggesting the Fe(II) is in high-spin configuration. The N $1s$ of benzimidazole can be divided into two peaks at 398.6 and 400.1 eV, which can be assigned to pyridinic N and pyrrolic N on the basis of the respective binding energies. Only one peak in the N $1s$ spectrum of $\text{Fe}_n(\text{bim})_{2n}$ was observed at 399.1 eV. This change can be attributed to the coordination of two nitrogen atoms of benzimidazole with Fe (II) atoms to form Fe-N bonds.

Supplementary Fig. 14 shows the FT-IR spectra of ferrocene, benzimidazole and $\text{Fe}_n(\text{bim})_{2n}$. It can be clearly seen that the characteristic bands of $3100\text{-}2600 \text{ cm}^{-1}$ assigned to N-H vibrations (Inorg. Chim. Acta 23, 155 (1977)) in benzimidazole disappears in the spectrum of $\text{Fe}_n(\text{bim})_{2n}$. It indicates the substitution of N-H in benzimidazole and formation of coordination bonds between Fe and N in $\text{Fe}_n(\text{bim})_{2n}$.

The results from XPS and FT-IR characterization confirmed the coordination reaction between ferrocene and benzimidazole, as well as the presence of Fe-N bonds in $\text{Fe}_n(\text{bim})_{2n}$.

Supplementary Figure 13. (a) Fe 2p XPS spectra of $\text{Fe}_n(\text{bim})_{2n}$ and ferrocene. (b) N 1s XPS spectra of $\text{Fe}_n(\text{bim})_{2n}$ and benzimidazole. XPS spectra were obtained using ESCALab250Xi. In the Fe 2p spectrum of ferrocene, the peaks at 707.8 and 720.6 eV are assigned to the binding energies of the $2p_{3/2}$ and $2p_{1/2}$ orbitals of Fe(II) species, respectively. A weak peak at 711.3 eV can be attributed to a satellite peak, an indicator of Fe(II) in low-spin configuration. The $2p_{3/2}$ and $2p_{1/2}$ orbitals of Fe(II) in the $\text{Fe}_n(\text{bim})_{2n}$ were observed at 710.5 and 723.9 eV respectively, shifting toward higher binding energies than those in the ferrocene. This can be explained by the decrease in electron cloud density around Fe atoms in $\text{Fe}_n(\text{bim})_{2n}$ compared to ferrocene, as benzimidazole has stronger electron-withdrawing ability than the cyclopentadienyl group of ferrocene. The satellite peaks for $\text{Fe}_n(\text{bim})_{2n}$ appear at 715.0 and 729.2 eV, suggesting the Fe(II) is in high-spin configuration¹. The N 1s of benzimidazole can be divided into two peaks at 398.6 and 400.1 eV, which can be assigned to pyridinic N and pyrrolic N on the basis of the respective binding energies. Only one peak in the N 1s spectrum of $\text{Fe}_n(\text{bim})_{2n}$ was observed at 399.1 eV. This change can be attributed to the coordination of two nitrogen atoms of benzimidazole with Fe (II) atoms to form Fe-N bonds. (The intensity normalization of XPS spectra was carried out.)

Supplementary Figure 14. FT-IR spectra of $\text{Fe}_n(\text{bim})_{2n}$, ferrocene and benzimidazole. FT-IR spectroscopy was recorded on VERTEX 70v. The characteristic bands of 3100-2600 cm^{-1} assigned to N-H vibrations² in benzimidazole disappears in the spectrum of $\text{Fe}_n(\text{bim})_{2n}$. It indicates the substitution of N-H in benzimidazole and formation of coordination bonds between Fe and N. (The intensity normalization of infrared spectra was carried out.)

Revisions made:

We have added the dose description of cryogenic TEM characterization in the capture of Figure 3.

“The cryogenic TEM was conducted under a low dose of 17.56 $\text{e}/\text{\AA}^2/\text{s}$.” (line 325-326)

We have added XPS and FT-IR characterizations in the main text and supporting information, with detailed descriptions in the legend of Supplementary Figs. 13 and 14.

“In addition, ... X-ray photoelectron spectroscopy (XPS, Supplementary Fig. 13) and Fourier transform infrared (FT-IR, Supplementary Fig. 14).” (line 168-172)

(5) The generality of the reported synthetic method is suggested to be demonstrated by other types of layered MOF structures.

ANS: Thank you for your advice. The SCA-CVD method has grown single crystalline 2D MOFs with different ligand molecules as shown in the main text. To further examine the generality of SCA-CVD, 2D MOFs with different metallic nodes were attempted to

be grown by SCA-CVD. As shown in Supplementary Fig. 4, the single crystals of 2D poly[Zn(benzimidazole)₂] have been successfully prepared, which indicates a potential for the generality of the SCA-CVD. It is worth noting that fulfilling a mature CVD preparation of a new material often takes several months or even years. As the revised time is limited, SCA-CVD growth of other metal-node MOFs and more types of layered MOFs will be conducted in future work.

For the sake of rigor, we removed the statement “These results suggest good versatility of our MOF-CVD method” from the main text.

Supplementary Figure 4 | The SCA-CVD growth of poly[Zn(benzimidazole)₂] flakes. (a) Crystal structure of a poly[Zn(benzimidazole)₂] single crystal viewed down the *c* axis (*a*–*b* plane). (b) Layered crystal structure of a poly[Zn(benzimidazole)₂] single crystal. (c) Typical optical image of poly[Zn(benzimidazole)₂] flakes. (d–g) STEM-EDS mapping of poly[Zn(benzimidazole)₂] flake. The metal source is bis(2,2,6,6-tetramethyl-3,5-heptanedionato) zinc(II).

Revisions made:

We have added “poly[Zn(benzimidazole)₂]” in the main text and Supplementary Fig. 4 in supporting information.

“Also, atomically thin MOF crystals, ... and poly[Zn(benzimidazole)₂] were successfully prepared via SCA-CVD (Supplementary Figs. 3 and 4)” (line 109-110)

We removed the statement “These results suggest good versatility of our MOF-CVD method” from the main text.

(6) Please give a summary and comparison Table to highlight the NH₃ sensing performance obtained by the ultrathin MOF/MoS₂ heterostructure.

ANS: Thank you very much for your advice. We have added a summary and comparison table on the NH₃ sensing performance of Fe_n(bim)_{2n}/MoS₂ heterostructures and other 19 materials in supporting information, as shown in Table S1.

Table S1. Summary and comparison of NH₃ sensing performance obtained with different materials.

Structure	Sensing environment	Response/ Recovery time (s)	LOD ^a (ppm)	Linearity range (ppm)	Amine selectivity	Ref.
Mesoporous carbons	RT ^b	120/240	1	—	NH ₃ , benzene, etc.	[3]
S-SWNT	RT	60-600/—	—	—	NH ₃ , NO ₂	[4]
Graphene aerogel	RT	100/500	0.01	0.02-85	NH ₃ , toluene, etc.	[5]
RGO	RT	~300-800/ ~1000-3500	—	200-2800	NH ₃ , methanol, etc.	[6]
Polyaniline-TiO ₂	RT	~35-45/ ~140-155	—	—	—	[7]
Ag	RT	180-600/—	—	1-10	—	[8]
CuO-MnO ₂	RT	120/600	—	—	—	[9]
ZnO Films	RT	~92-110/ ~111-113	5	—	NH ₃ , methanol, etc.	[10]
Silica modified CeO ₂	RT	32-760/ 141-2800	0.5	0.5-80	NH ₃ , H ₂ etc.	[11]
SnS ₂	RT	300/—	0.02	—	NH ₃ , H ₂ O ₂ , etc.	[12]
DNTT	RT	95/—	0.01	0.01-1	NH ₃ , H ₂ O ₂ , etc.	[13]
Cu ₃ HHTP ₂	RT	81.6/546.6	0.5	1-100	NH ₃ , CO, etc.	[14]
Cu ₃ (HHTP)(TH Q)	RT	99/154.2	0.02-0.35	1-~5x10 ⁵	NH ₃ , ethanol, etc.	[15]
Ti ₃ C ₂ T _x film	RT	—	0.13x10 ⁻³	—	NH ₃ , ethanol, etc.	[16]
WS ₂ nanoflakes	RT	~120/~150	—	1-5	NH ₃ , benzene, etc.	[17]
Few layered n-typed MoTe ₂	RT	—	~1	2-10	NH ₃ , NO ₂	[18]
Film of MoS ₂ powder	150 °C	400/~2200	—	10-50	NH ₃ , H ₂ , etc.	[19]
Au-dope MoS ₂ nanoflakes	90 °C	306/367	10	10-200	NH ₃ , toluene, etc.	[20]
Monolayer MoS ₂	RT	—	2.5	—	—	[21]
Fe _n (bim) _{2n} /MoS ₂	RT	9-60/80-200	0.5	1-100	NH ₃ , amines	This work

^a LOD, limit of detection, ^b RT, room temperature.

Revisions made:

We have added Table S1 in supporting information.

(7) The advantages and disadvantages of the proposed SCA-CVD method shall be compared with other existing ones. If possible, give some demonstrations to highlight the advantages, especially in the field of device applications.

ANS: Thank you for your professional advice.

To date, two major strategies have been employed to prepare 2D MOFs: one is top-down method (mechanical or liquid exfoliation from bulk crystals); the other is bottom-up method, such as interfacial synthesis, surfactant-assisted synthesis and CVD. Considerable efforts have been made in preparation of 2D MOFs. However, there are few reports on the preparation of independent monolayer single-crystal MOFs. The mechanical exfoliation method often involves small-size 2D product, low single-layer ratio and repeatability, and a lack of layer number controllability. In addition, due to the fragility of the bulk crystals of MOFs, it is more difficult to exfoliate MOFs crystals into monolayers compared to other 2D materials. In contrast, SCA-CVD method can afford a relatively high single-layer ratio and large-sized single-layer MOFs, contributing to more reliable sample preparation for applications.

The methods of interfacial synthesis and surfactant-assisted synthesis often face difficulties of low crystallinity and surface residual impurities. The SCA-CVD method can provide highly crystalline and clean 2D MOFs single crystals.

Regarding CVD methods for MOFs, big challenges remain. The reported MOFs grown by CVD are most limited to amorphous or polycrystalline films. SCA-CVD has achieved one-step preparation of large-sized monolayer MOFs single crystals with simple operations. Moreover, SCA-CVD method is an improvement of the traditional CVD method, which can meet the requirements of preparing MOFs. By modifying reaction conditions, 2D MOF single crystals with different ligands and metal nodes have been prepared by SCA-CVD. In addition, SCA-CVD demonstrates a good substrate generality. 2D MOFs single crystals have been grown on various substrates, even directly on other 2D materials.

Electron beam lithography technology is necessary for the fabrication of small-sized crystal device. However, damages to the samples especially for organic component samples are often inevitable due to the photoresist removal process. The large size in SCA-CVD grown $\text{Fe}_n(\text{bim})_{2n}$ single crystals enables the direct construction of electrodes using a hollow mask method without the use of electron beam lithography technology.

The directly grown MOFs on various substrates and 2D materials enabled direct device integration without transfer processes that often cause quality damage and contamination. As demonstrated by the ultrathin vdW heterostructure of $\text{Fe}_n(\text{bim})_{2n}/\text{MoS}_2$, atomically thin $\text{Fe}_n(\text{bim})_{2n}$ single crystals were directly grown onto

MoS₂ monolayers by SCA-CVD. The devices of Fe_n(bim)_{2n}/MoS₂ were obtained only by thermal evaporation gold electrodes on the top of the heterostructure with a mask. The properties of each component in the heterostructure can be maximally preserved.

Much progress in the preparation of atomically thin MOFs single crystals on the substrates have been made by SCA-CVD, however, current SCA-CVD method remain faces low production in mass (weight). The weight of products grown on a 1*1 cm² substrate is far lower than 100 μg due to the atomically thin nature of Fe_n(bim)_{2n}. Some applications that require a certain weight, such as catalysis and energy storage (lithium batteries, supercapacitors, composite materials, etc.), will be limited.

Revisions made:

We have added the discussion about the advantages of SCA-CVD in the main text.

“Compared with previous studies, SCA-CVD method demonstrates the following advantages: ... of good crystallinity.” (line 173-176)

“in which the transfer processes that often cause quality damage and contamination are not involved.” (line 214-215)

“The large size in SCA-CVD grown Fe_n(bim)_{2n} ... of electron beam lithography technology (Methods).” (line 220-223)

Reviewer #3 (Remarks to the Author):

1. The most critical problem is that the explanation of the growth of few layers is insufficient. The article explains the principle of reducing the kinetic limitations of lateral growth, but this process does not limit its vertical growth. Therefore, the principle of obtaining few layers or even single layers through this method is still unclear, and I hope the author can reasonably explain the reason of its vertical growth limiting effect.

ANS: Thank you for your professional advice. Actually, almost all reported two-dimensional materials prepared by CVD (such as graphene, TMDs) give priority to single-layer or few-layer growth, as long as they can achieve in-plane epitaxy. The growth of a 2D material is substantially affected by the high anisotropy in bonding. The $\text{Fe}_n(\text{bim})_{2n}$ is a typical Van der Waals layered crystal. Similar to most 2D materials, the atoms in a $\text{Fe}_n(\text{bim})_{2n}$ layer interact through strong chemical bonds, and the interactions between layers are weak van der Waals force (Nat. Chem. 10, 1001 (2018)). Density functional theory (DFT) shows that in most 2D materials, the strength of typical chemical bonds is between 2 and 8 eV per atom and the layer-layer interactions are 0.03-0.2 eV per atom. The strength of intralayer interaction is 1-2 orders of magnitude higher than that of interlayer interaction. A high bonding anisotropy is thus resulted in, in which intralayer bonding is prioritized over the layer-by-layer growth. Accordingly, in an equilibrium reversible thermodynamic process, selective growth of the single- or few-layer $\text{Fe}_n(\text{bim})_{2n}$ is favorable, as the case in CVD growth of most 2D Van der Waals crystals. (Refer to Chem. Rev. 121, 6321 (2021) and ACS Nano 17, 127, (2023))

Revisions made:

We have added the explanations for vertical growth limiting of $\text{Fe}_n(\text{bim})_{2n}$ in the main text.

“In addition, similar to most 2D materials...as the case in CVD growth of most 2D Van der Waals crystals.” (line 190-197)

2. Another question of concern is that the article mentioned that about 50% of the samples are monolayer. Whether a large-area view can be provided to show this result. In addition, can this ratio be regulated by growth conditions? How will the sample change as the growth time is shortened or extended?

ANS: Thank you for your question. We have added a large-area optical image of approximately 50% monolayer $\text{Fe}_n(\text{bim})_{2n}$ grown by SCA-CVD (Supplementary Fig.

1b). The growth conditions can affect this ratio. We found that this ratio often decreased when growth time exceeded 15 minutes, as shown in Supplementary Fig. 1c. In addition, the shorter the growth time, the smaller the average grain size of $\text{Fe}_n(\text{bim})_{2n}$, and the sparser the grain distribution. While the growth time extended, the average grain size of $\text{Fe}_n(\text{bim})_{2n}$ increased, accompanied by an increase in thickness and distribution density. (Supplementary Fig. 1a, c)

Supplementary Figure 1. Optical images of $\text{Fe}_n(\text{bim})_{2n}$ flakes with growth time of 5 minutes (a), 15 minutes (b) and 40 minutes (c). It can be seen that the shorter the growth time, the smaller the average grain size of $\text{Fe}_n(\text{bim})_{2n}$, and the sparser the grain distribution, and while the growth time extended, the average grain size of $\text{Fe}_n(\text{bim})_{2n}$ increased, accompanied by an increase in thickness and distribution density.

Revisions made:

We have added Supplementary Fig. 1 in supporting information.

3. The article mentioned that the SCA-CVD method has good versatility, but the listed MOF crystals that can be prepared by this method are similar, except for the substituents of benzimidazole. Therefore, whether the method can be described as versatile remains to be reconsidered.

ANS: Thank you for your suggestion. The SCA-CVD method has grown single crystalline 2D MOFs with different ligand molecules as shown in the main text. To further examine the generality of SCA-CVD, 2D MOFs with different metallic nodes were attempted to be grown by SCA-CVD. As shown in Supplementary Fig. 4, the single crystals of 2D poly[$\text{Zn}(\text{benzimidazole})_2$] have been successfully prepared, which indicates a potential for the generality of the SCA-CVD. It is worth noting that fulfilling a mature CVD preparation of a new material often takes several months or even years. As the revised time is limited, SCA-CVD growth of other metal-node MOFs and more types of layered MOFs will be conducted in future work.

For the sake of rigor, we removed the statement “These results suggest good versatility of our MOF-CVD method” from the main text.

Supplementary Figure 4 | The SCA-CVD growth of poly[Zn(benzimidazole)₂] flakes. (a) Crystal structure of a poly[Zn(benzimidazole)₂] single crystal viewed down the *c* axis (*a*–*b* plane). (b) Layered crystal structure of a poly[Zn(benzimidazole)₂] single crystal. (c) Typical optical image of poly[Zn(benzimidazole)₂] flakes. (d–g) STEM-EDS mapping of poly[Zn(benzimidazole)₂] flake. The metal source is bis(2,2,6,6-tetramethyl-3,5-heptanedionato) zinc(II).

Revisions made:

We have added “poly[Zn(benzimidazole)₂]” in the main text and Supplementary Fig. 4 in supporting information.

“Also, atomically thin MOF crystals, ... and poly[Zn(benzimidazole)₂] were successfully prepared via SCA-CVD (Supplementary Figs. 3 and 4)” (line 109-110)

We removed the statement “These results suggest good versatility of our MOF-CVD method” from the main text.

4. In the AFM image of Figure 1f, there are obvious protrusion points. What is its source and whether it will affect the quality of the material?

ANS: Thank you for your question. The protrusion points were caused by the absorption of dust in ambient environment. These dust will not affect quality of the material. The presence of protrusion points is very common in AFM images, such as graphene (Fig. R8a), CrSe₂ (Fig. R8b), SnS₂ (Fig. R8c), Bi₂O₂Se (Fig. R8d), MoS₂ (Fig. R8e) and FeS₂ (Fig. R8f), etc. and no damage in quality has been reported.

Moreover, this problem can be greatly improved when the samples are exposed in ambient environment as little as possible followed by conducted AFM measurement in the clean room. As shown in Fig. R9, an AFM image of Fe_n(bim)_{2n} with almost no protrusion points has been obtained.

Figure R8. The presence of protrusion points is very common in AFM images. (a) AFM image of the as-grown graphene on Ge(110) substrate. (Science 344, 286–289 (2014)) (b) AFM image of the CrSe₂ nanosheet. (Nat. Mater. 20, 818–825 (2021)) (c) AFM image of the SnS₂ flake. (Nat. Nanotechnol. 18, 448–455 (2023)) (d) AFM image of monolayer Bi₂O₂Se. (Nat. Nanotechnol. 12, 530–534 (2017)) (e) Optical image and AFM image (inset) of 1T' MoS₂ monolayer flakes grown on mica. (Nat. Mater. 17, 1108–1114 (2018)) (f) AFM image of FeS₂ flake. (Nat. Mater. 22, 450–458 (2023))

Figure R9. AFM image of $\text{Fe}_n(\text{bim})_{2n}$. The $\text{Fe}_n(\text{bim})_{2n}$ was first preserved in vacuum and measured in the clean room.

5. The article mentioned the high selectivity of the material to NH_3 . I wonder how stable the selectivity is this time and whether it can still be maintained after multiple cycles.

ANS: Thanks a lot for your question. To test the reproducibility of NH_3 selectivity of the $\text{Fe}_n(\text{bim})_{2n}/\text{MoS}_2$ sensor, we exposed the sensor to NH_3 (20 ppm) and tert-butylamine (20 ppm) alternately, in which the injection time was 30s and the interval time was 90s. As shown in Supplementary Fig. 17, exposure to NH_3 resulted in rapid responses, while almost no response was observed by the introduction of tert-butylamine. Such selective NH_3 response was maintained even after 30 successive cycles of alternating exposure to NH_3 and tert-butylamine, which indicates the good reproducibility of the NH_3 selectivity of the $\text{Fe}_n(\text{bim})_{2n}/\text{MoS}_2$ sensor.

Supplementary Figure 17. Response of the $\text{Fe}_n(\text{bim})_{2n}/\text{MoS}_2$ sensor during 30 successive cycles of exposure to NH_3 (20 ppm) and tert-butylamine (20 ppm) alternately. The red arrow represents the injection of tert-butylamine vapour.

Revisions made:

We have added the description of the NH_3 selectivity stability of the $\text{Fe}_n(\text{bim})_{2n}/\text{MoS}_2$ sensor in the main text and Supplementary Fig. 17 in supporting information.

“In addition, the NH_3 selectivity of the $\text{Fe}_n(\text{bim})_{2n}/\text{MoS}_2$ sensor has a good reproducibility (Supplementary Fig. 17).” (line 256-257)

6. In addition, the article's explanation for its precise gate effect is that it has no undesired gas permeation pathways. Does this permeation pathway only take effect for ammonia gas? What is the result for small molecules of the gas (such as benzene)?

ANS: Thank you for your question. We analyzed the pore size of $\text{Fe}_n(\text{bim})_{2n}$ to make the mechanism of gas selectivity clearer. According to the crystallographic data, the theoretical aperture size of $\text{Fe}_n(\text{bim})_{2n}$ unit is estimated to be about 0.21 nm, as shown in Supplementary Fig. 18. Considering the structure flexibility of MOF owing to the aromatic rings flip-flop, the effective pore size should be slightly larger than the theoretical aperture size. (Angew. Chem. Int. Ed. 62, e2023129 (2023); Science 346, 1356 (2014)) Organic amine vapours including tert-butylamine, tert-pentylamine, isopropylamine and 2-butylamine all have kinetic diameters far exceeding the effective aperture size of $\text{Fe}_n(\text{bim})_{2n}$. As expected, due to size exclusion effect by the small

aperture size of $\text{Fe}_n(\text{bim})_{2n}$, the gate effect of the $\text{Fe}_n(\text{bim})_{2n}$ layer prevents these organic amine vapours from accessing the surface of MoS_2 , causing almost no gas sensing response. While NH_3 molecules with a kinetic diameter of approximately 2.6 Å which is comparable to the effective pore aperture of $\text{Fe}_n(\text{bim})_{2n}$, can easily pass through the $\text{Fe}_n(\text{bim})_{2n}$ network and reach the MoS_2 surface, resulting in gas responses. Moreover, the atomic-level thickness of $\text{Fe}_n(\text{bim})_{2n}$ ensured the retention of the good sensitivity of MoS_2 monolayers in the heterostructure. Finally, a highly selective ammonia sensing was realized in the ultrathin vdW heterostructure of $\text{Fe}_n(\text{bim})_{2n}/\text{MoS}_2$.

Based on the above analysis, the gate effect of the $\text{Fe}_n(\text{bim})_{2n}$ layer not only takes effect for NH_3 , but for analysts with kinetic diameters that within the range of the effective aperture size of the $\text{Fe}_n(\text{bim})_{2n}$. The benzene molecule has a kinetic diameter of 5.85 Å (Pure Appl. Chem. 72, 2289 (2000)), much larger than the effective aperture size of the MOF, so benzene molecules failed to pass through the $\text{Fe}_n(\text{bim})_{2n}$. As demonstrated experimentally, exposure to benzene caused obvious response in pure MoS_2 sensor while almost no response in the $\text{Fe}_n(\text{bim})_{2n}/\text{MoS}_2$ sensor. (Fig. R10)

Supplementary Figure 18. 3D crystalline structure showing the pore of the $\text{Fe}_n(\text{bim})_{2n}$ unit. Fe, N, C and H atoms are shown in orange, purple, gray and white balls, respectively.

Figure R10. Sensing responses of a monolayer MoS₂ (black line) and a Fe_n(bim)_{2n}/MoS₂ (red line) when exposed to benzene vapour (50 ppm).

Revisions made:

The clear elaboration of the gas selectivity mechanism has been added in the main text.

“According to the crystallographic data, ... can easily pass through the Fe_n(bim)_{2n} network and reach the MoS₂ surface, resulting in gas responses.” (line 262-273)

Supplementary Fig. 18 was added in supporting information.

7. A small issue. The HRAFM results in Figure 3c are not clear enough. FFT processing can be considered.

ANS: Thanks for your advice. The FFT processed HRAFM image was added in Supplementary Fig. 11.

Supplementary Figure 11. HRAFM image of Fe_n(bim)_{2n} after FFT processing of Fig. 3c.

Revisions made:

We have added Supplementary Fig. 11 in supporting information.

Reviewer #4 (Remarks to the Author):

1. Would it be possible to use GIWAXS measurements to ensure that the crystal orientation of the MOF ($\text{Fe}_n(\text{bim})_{2n}$) on different substrates are all single crystals?

ANS: Thank you very much for your advice. We have tried our best to perform GIWAXS characterization for the $\text{Fe}_n(\text{bim})_{2n}$ on SiO_2/Si substrates. Two GIWAXS devices, XEUSS and XEUSS 2.0, were used respectively. But no signals have been detected on the $\text{Fe}_n(\text{bim})_{2n}$ flakes (Fig. R1). The atomically thin nature of the $\text{Fe}_n(\text{bim})_{2n}$ is supposed to be responsible for this result. We noted that utilizing GIWAXS technique to characterize atomically thin crystals is extremely rare and has been rarely reported in the field of 2D materials. The GIWAXS measurement is highly likely not suitable for characterizing 2D materials with atomic thickness. To further verify this, GIWAXS measurements were performed on wafer-scale monolayer single-crystal graphene and large-scale monolayer single-crystal MoS_2 using the same equipment. As in the case of $\text{Fe}_n(\text{bim})_{2n}$, no detectable signals were observed in both graphene and MoS_2 (Fig. R1). One of the main reasons for this result may be the energy of the light source used in the GIWAXS measurement is low for atomically thin 2D materials to produce detectable signals.

Referring to a large number of references, we found the method of multi-point and multi-region electron diffraction is commonly used to validate whether large-sized grains of atomically thin materials are single crystals (Fig. R2, Nat. Mater. 15, 43 (2016); Nat. Commun. 4, 2096 (2013); Nat. Nanotechnol. 15, 289 (2020); Nat. Commun. 6, 8569 (2015); Proc. Natl. Acad. Sci. USA 109, 7992 (2012)). Therefore, we carefully performed selective area electron diffraction (SAED) on nine different domains of an individual $\text{Fe}_n(\text{bim})_{2n}$ flake grown on a KBr substrate. As shown in Supplementary Fig. 9, only one set of quasi-four-fold symmetric diffraction spots was observed in each domain, indicating that these domains are single crystals. In addition, the lattice orientations extracted from the SAED patterns in the nine domains were found almost identical. This result of single crystals with the same lattice orientation in the domains confirms that the entire rectangle flake of $\text{Fe}_n(\text{bim})_{2n}$ is a single crystal. Notably, all the individual $\text{Fe}_n(\text{bim})_{2n}$ flakes we have grown are regular rectangles, which matches well with the monoclinic crystal cell structure of $\text{Fe}_n(\text{bim})_{2n}$. Generally, crystals with regular geometric shapes that match their cell structure are considered single crystals, that is, an individual rectangular $\text{Fe}_n(\text{bim})_{2n}$ flake is a single crystal, as confirmed by the result of multi-point and multi-region SAED. In order to further validate the conclusion, individual $\text{Fe}_n(\text{bim})_{2n}$ flakes in rectangle grown on SiO_2/Si , sapphire, quartz and SiN substrates respectively were examined using the multi-point and multi-region electron diffraction method. As shown in Supplementary Fig. 10, one set of diffraction spots and the same lattice orientation were demonstrated in different regions of the same rectangular flake, indicating all these individual flakes are single crystals. Therefore, the individual rectangular flakes of $\text{Fe}_n(\text{bim})_{2n}$ grown on different substrates are all single crystals.

Figure R1. GIWAXS image of Fe_n(bim)_{2n} nanosheets (a), wafer-scale monolayer single-crystal graphene (b) and large-scale monolayer single-crystal MoS₂ (c).

Figure R2. For 2D materials, conducting SAED measurements at different locations on an individual large-area flake to demonstrate that the flake is a large-sized single crystal is a widely recognized method. Here we listed the relevant literatures. (a) TEM/SAED investigations showing that the graphene prepared by local feeding method is mono-layered and single crystalline. (Nat. Mater. 15, 43–47 (2016)) (b) Structural characterization of large single crystals of monolayer graphene. (Nat. Commun. 4, 2096 (2013)) (c) TEM characterization of a bilayer graphene film. (Nat. Nanotechnol. 15, 289–295 (2020)) (d) Structural characterization of monolayer single-crystal WS₂ domains. (Nat. Commun. 6, 8569 (2015)) (e) Raman and TEM characterizations of single-crystalline hexagonal graphene flakes. (Proc. Natl. Acad. Sci. USA 109, 7992–7996 (2012))

Supplementary Figure 9. (a) Low-magnification TEM image of an individual $\text{Fe}_n(\text{bim})_{2n}$ flake. (b-j) SAED patterns in the area enclosed by green circles in a. Only one set of quasi-four-fold symmetric diffraction spots was observed in each area, indicating that these areas are single crystals. The lattice orientations extracted from the SAED patterns in the nine areas were found almost identical. The above results confirm that the entire rectangle flake of $\text{Fe}_n(\text{bim})_{2n}$ is a single crystal.

Supplementary Figure 10 | TEM characterizations of $\text{Fe}_n(\text{bim})_{2n}$ flakes grown on different substrates. TEM images and SAED patterns at four randomly selected areas (green circles) of $\text{Fe}_n(\text{bim})_{2n}$ flakes grown on SiO_2/Si (a), sapphire (b), quartz (c) and SiN (d). One set of diffraction spots and the same lattice orientation were demonstrated in different regions of the same rectangular flake, indicating all these individual flakes are single crystals. Therefore, the individual rectangular flakes of $\text{Fe}_n(\text{bim})_{2n}$ grown on different substrates are all single crystals.

Revisions made:

We have added the description of multi-point and multi-region SAED characterization in the main text.

“An individual $\text{Fe}_n(\text{bim})_{2n}$ flake in rectangle shape has been further confirmed as a single crystal by collecting SAED patterns at different positions throughout the entire flake (Supplementary Figs. 9 and 10).” (line 153-155)

Supplementary Figs. 9 and 10 were added in supporting information, with the detailed descriptions in their figure captures.

2. It would be decent to elaborate on more details about the MOF ($\text{Fe}_n(\text{bim})_{2n}$), such as XPS, FTIR, NMR, and TGA. Furthermore, some references to Figures 1a and 1b could be enclosed since insufficient details characterize that MOF.

ANS: Thank you for your professional advice. We have carried out XPS and FT-IR measurements to reveal more information on the $\text{Fe}_n(\text{bim})_{2n}$.

Supplementary Fig. 13 displays the Fe(II) $2p$ core-level and N $1s$ core-level XPS spectra of $\text{Fe}_n(\text{bim})_{2n}$, ferrocene and benzimidazole. In the Fe $2p$ spectrum of ferrocene, the peaks at 707.8 and 720.6 eV are assigned to the binding energies of the $2p_{3/2}$ and $2p_{1/2}$ orbitals of Fe(II) species, respectively. A weak peak at 711.3 eV can be attributed to a satellite peak, an indicator of Fe(II) in low-spin configuration. The $2p_{3/2}$ and $2p_{1/2}$ orbitals of Fe(II) in the $\text{Fe}_n(\text{bim})_{2n}$ were observed at 710.5 and 723.9 eV respectively, shifting toward higher binding energies than those in the ferrocene. This can be explained by the decrease in electron cloud density around Fe atoms in $\text{Fe}_n(\text{bim})_{2n}$ compared to ferrocene, as benzimidazole has stronger electron-withdrawing ability than the cyclopentadienyl group of ferrocene. The satellite peaks for $\text{Fe}_n(\text{bim})_{2n}$ appear at 715.0 and 729.2 eV, suggesting the Fe(II) is in high-spin configuration. The N $1s$ of benzimidazole can be divided into two peaks at 398.6 and 400.1 eV, which can be assigned to pyridinic N and pyrrolic N on the basis of the respective binding energies. Only one peak in the N $1s$ spectrum of $\text{Fe}_n(\text{bim})_{2n}$ was observed at 399.1 eV. This change can be attributed to the coordination of two nitrogen atoms of benzimidazole with Fe (II) atoms to form Fe-N bonds.

Supplementary Fig. 14 shows the FT-IR spectra of ferrocene, benzimidazole and $\text{Fe}_n(\text{bim})_{2n}$. It can be clearly seen that the characteristic bands of 3100-2600 cm^{-1} assigned to N-H vibrations (Inorg. Chim. Acta 23, 155 (1977)) in benzimidazole disappeared in the spectrum of $\text{Fe}_n(\text{bim})_{2n}$. It indicates the substitution of N-H in benzimidazole and formation of coordination bonds between Fe and N.

The results from XPS and FT-IR characterization confirmed the coordination reaction between ferrocene and benzimidazole, as well as the presence of Fe-N bonds in $\text{Fe}_n(\text{bim})_{2n}$.

It is worth noting that both TGA and NMR measurements require the test sample with

a certain weight. The $\text{Fe}_n(\text{bim})_{2n}$ flakes were grown on substrates and have a very low mass due to their atomically thin nature. The weight of $\text{Fe}_n(\text{bim})_{2n}$ samples grown on a $1 \times 1 \text{ cm}^2$ substrate is far lower than $100 \mu\text{g}$. Therefore, it is very hard to use TGA or NMR to characterize the $\text{Fe}_n(\text{bim})_{2n}$ flakes.

We have enclosed reference 10 in the description section of Fig. 1a and 1b.

Supplementary Figure 13. (a) Fe 2p XPS spectra of $\text{Fe}_n(\text{bim})_{2n}$ and ferrocene. (b) N 1s XPS spectra of $\text{Fe}_n(\text{bim})_{2n}$ and benzimidazole. XPS spectra were obtained using ESCALab250Xi. In the Fe 2p spectrum of ferrocene, the peaks at 707.8 and 720.6 eV are assigned to the binding energies of the $2p_{3/2}$ and $2p_{1/2}$ orbitals of Fe(II) species, respectively. A weak peak at 711.3 eV can be attributed to a satellite peak, an indicator of Fe(II) in low-spin configuration. The $2p_{3/2}$ and $2p_{1/2}$ orbitals of Fe(II) in the $\text{Fe}_n(\text{bim})_{2n}$ were observed at 710.5 and 723.9 eV respectively, shifting toward higher binding energies than those in the ferrocene. This can be explained by the decrease in electron cloud density around Fe atoms in $\text{Fe}_n(\text{bim})_{2n}$ compared to ferrocene, as benzimidazole has stronger electron-withdrawing ability than the cyclopentadienyl group of ferrocene. The satellite peaks for $\text{Fe}_n(\text{bim})_{2n}$ appear at 715.0 and 729.2 eV, suggesting the Fe(II) is in high-spin configuration¹. The N 1s of benzimidazole can be divided into two peaks at 398.6 and 400.1 eV, which can be assigned to pyridinic N and pyrrolic N on the basis of the respective binding energies. Only one peak in the N 1s spectrum of $\text{Fe}_n(\text{bim})_{2n}$ was observed at 399.1 eV. This change can be attributed to the coordination of two nitrogen atoms of benzimidazole with Fe (II) atoms to form Fe-N bonds. (The intensity normalization of XPS spectra was carried out.)

Supplementary Figure 14. FT-IR spectra of $\text{Fe}_n(\text{bim})_{2n}$, ferrocene and benzimidazole. FT-IR spectroscopy was recorded on VERTEX 70v. The characteristic bands of 3100-2600 cm^{-1} assigned to N-H vibrations² in benzimidazole disappeared in the spectrum of $\text{Fe}_n(\text{bim})_{2n}$. It indicates the substitution of N-H in benzimidazole and formation of coordination bonds between Fe and N. (The intensity normalization of infrared spectra was carried out.)

Revisions made:

We have added XPS and FT-IR characterizations in the main text and supporting information, with detailed descriptions in the legend of Supplementary Figs. 13 and 14.

“In addition, ... X-ray photoelectron spectroscopy (XPS, Supplementary Fig. 13) and Fourier transform infrared (FT-IR, Supplementary Fig. 14).” (line 168-172)

We have enclosed reference 10 in the legend of Fig. 1a, b.

“...omitted for clarity. (ref. 10) **b**, Layered crystal structure of a $\text{Fe}_n(\text{bim})_{2n}$ single crystal viewed along the *b* axis. (ref. 10)” (line 302-303)

3. The reference for supporting the properties of $\text{Fe}_n(\text{bim})_{2n}$ would be insufficient since the MOF in that cited article is based on zinc metal. Furthermore, the fabrication process was a top-down process, but not CVD. Therefore, sharing all the physical and chemical properties would fall short of being persuasive. It would be better to analyze pore size distribution and the mechanism of gas selectivity.

ANS: Thank you very much for your suggestion. Because the weight of $\text{Fe}_n(\text{bim})_{2n}$ samples grown on a $1 \times 1 \text{ cm}^2$ substrate is far lower than 100 μg , it is very hard to

experimentally measure the pore size distribution. Therefore, we theoretically analyzed the pore size of $\text{Fe}_n(\text{bim})_{2n}$ to gain a deeper understanding of the mechanism of gas selectivity. According to the crystallographic data, the theoretical aperture size of $\text{Fe}_n(\text{bim})_{2n}$ unit is estimated to be about 0.21 nm, as shown in Supplementary Fig. 18. Considering the structure flexibility of MOF owing to the aromatic rings flip-flop, the effective pore size should be slightly larger than the theoretical aperture size. (Angew. Chem. Int. Ed. 62, e2023129 (2023); Science 346, 1356 (2014)) Organic amine vapours including tert-butylamine, tert-pentylamine, isopropylamine and 2-butylamine all have kinetic diameters far exceeding effective aperture size of $\text{Fe}_n(\text{bim})_{2n}$. As expected, due to size exclusion effect by the small aperture size of $\text{Fe}_n(\text{bim})_{2n}$, the gate effect of the $\text{Fe}_n(\text{bim})_{2n}$ layer prevents these organic amine vapours from accessing the surface of MoS_2 , causing almost no gas sensing response. While NH_3 molecules with a kinetic diameter of approximately 2.6 Å which is comparable to the effective pore aperture of $\text{Fe}_n(\text{bim})_{2n}$, can easily pass through the $\text{Fe}_n(\text{bim})_{2n}$ network and reach the MoS_2 surface, resulting in gas responses. Moreover, the atomic-level thickness of $\text{Fe}_n(\text{bim})_{2n}$ ensured the retention of the good sensitivity of MoS_2 monolayers in the heterostructure. Finally, a highly selective ammonia sensing was realized in the ultrathin vdW heterostructure of $\text{Fe}_n(\text{bim})_{2n}/\text{MoS}_2$.

Supplementary Figure 18. 3D crystalline structure showing the pore of the $\text{Fe}_n(\text{bim})_{2n}$ unit. Fe, N, C and H atoms are shown in orange, purple, gray and white balls, respectively.

Revisions made:

The clear elaboration of pore size and the gas selectivity mechanism has been added in the main text.

“According to the crystallographic data, ... can easily pass through the $\text{Fe}_n(\text{bim})_{2n}$ network and reach the MoS_2 surface, resulting in gas responses.” (line 262-273)

Supplementary Fig. 18 was added in supporting information.

4. What is the design of the gas flow orientation inside the experimental apparatus that causes the metal precursor and ligand to flow in opposite directions?

ANS: Thank you for your question. The gas flow orientation in the SCA-CVD growth was designed as shown in Fig. R11. A quartz inner tube with one end open and one end sealed was used, in which benzimidazole and ferrocene were respectively loaded at the two ends. High purity nitrogen as the carrier gas was used to prevent oxygen in atmosphere from entering the CVD system and maintain an inert atmosphere. The carrier gas flowed from the upstream of the outer large quartz tube (near the sealed end of the inner tube) to the downstream of the outer quartz tube throughout the CVD process. The carrier gas flow was set at a small rate of 5 sccm, which had almost no impact on the diffusion of metal precursor and ligand in the one-ended sealed inner tube. The diffusion of metal precursor and ligand in the inner tube was mainly driven by their concentration gradient. In addition, the sealed end of the inner tube is located upstream of the gas flow, which makes the reverse diffusion (to the benzimidazole) of ferrocene vapour much easier inside the inner tube.

Figure R11. Schematics of the SCA-CVD growth.

5. Would it be possible to describe how to get the yield of MOF ($\text{Fe}_n(\text{bim})_{2n}$) in as much detail as possible?

ANS: Thanks for your question. We used AFM measurement to count the numbers of single-layer and few-layer to obtain the monolayer ratio. The theoretical thickness of a monolayer is 0.938 nm as estimated from crystallographic data. Noting that 2D materials are often raised by a few angstroms above the supporting surface, the $\text{Fe}_n(\text{bim})_{2n}$ flakes with thickness below 1.4 nm is considered as monolayers. In addition, we think that it is more reasonable to describe "monolayer ratio" than "monolayer yield" in the CVD method. For the sake of rigor, we have changed the statement "the yield of the monolayer was approximately 50%" to "the ratio of the monolayer was approximately 50%" in the manuscript.

Revisions made:

We have revised the statement that "the yield of the monolayer was approximately 50%" to "the ratio of the monolayer was approximately 50%" in the main text, line 98.

6. What would be the criteria for determining whether monolayer single-crystals of $\text{Fe}_n(\text{bim})_{2n}$ up to $42\ \mu\text{m}$ in grain sizes regarding your research? The optical image of Figure 1c cannot be evident to prove that the grain size is $42\ \mu\text{m}$; however, in Figure 3a, the observed crystal size is only $6.5\ \mu\text{m}$ and seems poly-crystal. It could be more convincing with the elaboration.

ANS: Thank you for your question. As the individual $\text{Fe}_n(\text{bim})_{2n}$ flakes are shaped in rectangle, we defined the diagonal length of the rectangle as the crystal size of the $\text{Fe}_n(\text{bim})_{2n}$ flake. In order to make readers more clearly and intuitively get the grain size, we defined side length as the crystal size. Moreover, in the past few months, we have obtained monolayer $\text{Fe}_n(\text{bim})_{2n}$ with sizes exceeding $42\ \mu\text{m}$, as show in Supplementary Fig. 1b.

Owing to the atomically thin nature, monolayer $\text{Fe}_n(\text{bim})_{2n}$ has very low contrast, almost transparent in the field of view of TEM. A high magnification is favorable for clear observation during TEM measurement. However, to image an entire large-sized $\text{Fe}_n(\text{bim})_{2n}$, a lower magnification is usually required to obtain a larger field of view, in which the morphology of the sample often cannot be clearly observed. Therefore, to clearly image an entire crystal, small-sized $\text{Fe}_n(\text{bim})_{2n}$ crystals are preferred for TEM observation.

We carefully examined the SAED pattern in Fig. 3a. There was only one set of diffraction patterns confirming the observed crystal was a single crystal. The stretching of some diffraction points was observed in the pattern, which is not rare in SAED characterization for single crystals. As demonstrated in Fig. R12, diffraction point stretching was also observed in the SAED patterns of MOF single crystals (Nat. Chem. 10, 1001 (2018); Science 346, 1356 (2014)). Moreover, a polycrystalline structure features multiple sets or even circular diffraction pattern, clearly different from the SAED pattern in Fig. 3a. So, the observed crystal is a single crystal.

Supplementary Figure 1b. Optical images of $\text{Fe}_n(\text{bim})_{2n}$ flakes with growth time of 15 minutes.

Figure R12. Stretching of diffraction points is common phenomenon in SEAD measurements. Here we listed the relevant literatures. (a) SAED pattern of MUV-1-H. (Nat. Chem. 10, 1001–1007 (2018)) (b) SAED pattern of $Zn_2(bim)_4$. (Science 346, 1356–1359 (2014))

Revisions made:

We have revised the crystal sizes and updated the maximum size of the monolayer $Fe_n(bim)_{2n}$ as $62 \mu m$ in the main text, line 101.

“with lengths for the monolayer and the few-layer up to $62 \mu m$ and $105 \mu m$, respectively”

7. As mentioned in the manuscript, the as-synthesized MOF characterized as single-crystal was grown on the KBr substrate. Nonetheless, the applied device was the one that was grown on MoS_2 . It would be more decent to illustrate that the MOF on MoS_2 was also single-crystal.

ANS: Thank you for your advice. The individual $Fe_n(bim)_{2n}$ flakes grown onto the monolayer MoS_2 are rectangles in shape, as same as that on the other substrates (such as KBr, SiO_2/Si). To validate the single crystal nature of these rectangular $Fe_n(bim)_{2n}$ on MoS_2 , SAED measurements were conducted on randomly selected three different regions in the area of $Fe_n(bim)_{2n}/MoS_2$ heterostructure, as shown in Supplementary Fig. 15. The three regions all demonstrated two different sets of diffraction patterns, one set of six-fold diffraction patterns (yellow circle) belonging to MoS_2 and one set of quasi-four-fold diffraction patterns (red circle) belonging to $Fe_n(bim)_{2n}$. This suggests all the three regions are composed of single-crystalline $Fe_n(bim)_{2n}$ stacked onto the single-crystalline MoS_2 . Comparing the quasi-four-fold diffraction patterns in the three regions, the same direction of arranged diffraction spots was found, which suggests $Fe_n(bim)_{2n}$ in the three regions share the same lattice orientation. Therefore, the entire $Fe_n(bim)_{2n}$ flake on MoS_2 is single-crystal.

Supplementary Figure 15 | TEM characterizations of Fe_n(bim)_{2n}/MoS₂ heterostructures. (a) Low-magnification TEM image of Fe_n(bim)_{2n}/MoS₂ heterostructure. (b-d) SAED patterns for three regions indicated by green circles. The three regions all demonstrated two different sets of diffraction patterns, one set of six-fold diffraction patterns (yellow circle) belonging to MoS₂ and one set of quasi-four-fold diffraction patterns (red circle) belonging to Fe_n(bim)_{2n}. This suggests all the three regions are composed of single-crystalline Fe_n(bim)_{2n} stacked onto the single-crystalline MoS₂. Comparing the quasi-four-fold diffraction patterns of the three regions, the same direction of arranged diffraction spots was found, which suggests Fe_n(bim)_{2n} in the three regions share the same lattice orientation. Therefore, the entire Fe_n(bim)_{2n} flake on MoS₂ is single-crystal.

Revisions made:

We have added Supplementary Fig. 15 in supporting information, with the detailed descriptions in the figure capture.

REVIEWER COMMENTS

Reviewer #1 (Remarks to the Author):

I am generally happy with the replies from the authors. Much further efforts have been clearly made to enhance the structure characterizations. I have two minor final comments on the replies:

1. The authors replied that "the lattice orientations extracted from the SAED patterns in the nine domains were found almost identical" How far away are these domains? That is critical to claim large single crystal domains from the paper.

2. About electronic structure "This behavior indicates the $\text{Fen}(\text{bim})_2\text{n}$ is likely an insulator with a large bandgap. To further determine the electronic property, field effect transistors were fabricated on the $\text{Fen}(\text{bim})_2\text{n}$, and found that the electric current of $\text{Fen}(\text{bim})_2\text{n}$ transistors was below the low detection limit of 10-12A. This suggests the $\text{Fen}(\text{bim})_2\text{n}$ is an insulator. The absorption spectroscopy of $\text{Fen}(\text{bim})_2\text{n}$ showed no visible range absorption." I reached an opposite conclusion by checking the data in Figure R3b: It is not that you do not have absorption in the UV regime; the absorption is just weak in the entire regime. The material has clear absorption inside the UV regime. The absorption at 800 nm is still non-zero, which may indicate a small bandgap of $\text{Fen}(\text{bim})_2\text{n}$.

Reviewer #2 (Remarks to the Author):

I am satisfied with the revisions that the authors have made. The manuscript can be accepted for publication as it is.

Reviewer #3 (Remarks to the Author):

The author solved my problem very well. The current article has rich data and reliable conclusions. I think the article can be published in Nature Communications after a minor revision.

Some new data graphics may be updated into text graphics, such as GIWAXS.

Theoretical calculations related to in-plane extension, including DFT, need to be described in detail. The optical microscope image of the sensor device needs to be given in SI.

Reviewer #4 (Remarks to the Author):

The definition of a single crystal is a crystalline solid for which the periodic and repeated atomic pattern extends throughout its entirety without interruption. (Materials Science and Engineering: An Introduction, 10th Edition, William D. Callister Jr., David G. Rethwisch)

The grain boundaries in Figure 3 a and b are obvious, which means that there are a lot of grains growth on the substrate and the crystal structure is not continuous. Despite those grains with the same crystal orientation, they cannot be called single crystals. Therefore, the optical image of Supplementary Figure 1b may not be evident to prove the grain size is 62 μm .

RESPONSE TO REVIEWERS' COMMENTS

Reviewer #1 (Remarks to the Author):

(1) The authors replied that "the lattice orientations extracted from the SAED patterns in the nine domains were found almost identical" How far away are these domains? That is critical to claim large single crystal domains from the paper.

ANS: Thank you very much for your suggestion. In Supplementary Fig. 9, the distance between adjacent domains is measured to be 3.7~ 4.4 μm and the size of the $\text{Fe}_n(\text{bim})_{2n}$ flake is 15.1 μm *13.8 μm . The SAED for nine domains are large enough to exhibit the characteristic of the flake. Therefore, the SAED result that these domains are single crystals with identical lattice orientation can confirm the entire flake containing these domains is a single crystal.

Revisions made:

The distance between adjacent domains was added in the legend of Supplementary Fig. 9 in supporting information.

Supplementary Figure 9. (a) Low-magnification TEM image of an individual $\text{Fe}_n(\text{bim})_{2n}$ flake. (b-j) SAED patterns of nine domains enclosed by green circles in a. The distance between adjacent domains is measured to be 3.7~ 4.4 μm . Only one set of quasi-four-fold symmetric diffraction spots was observed in each area, indicating that these areas are single crystals. The lattice orientations extracted from the SAED patterns in the nine areas were found almost identical. The above results confirm that the entire rectangle flake of $\text{Fe}_n(\text{bim})_{2n}$ is a single crystal.

(2) About electronic structure "This behavior indicates the $\text{Fe}_n(\text{bim})_{2n}$ is likely an insulator with a large bandgap. To further determine the electronic property, field effect transistors were fabricated on the $\text{Fe}_n(\text{bim})_{2n}$, and found that the electric current of $\text{Fe}_n(\text{bim})_{2n}$ transistors was below the low detection limit of 10^{-12}A . This suggests the $\text{Fe}_n(\text{bim})_{2n}$ is an insulator. The absorption spectroscopy of $\text{Fe}_n(\text{bim})_{2n}$ showed no

visible range absorption." I reached an opposite conclusion by checking the data in Figure R3b: It is not that you do not have absorption in the UV regime; the absorption is just weak in the entire regime. The material has clear absorption inside the UV regime. The absorption at 800 nm is still non-zero, which may indicate a small bandgap of $\text{Fe}_n(\text{bim})_{2n}$.

ANS: Thank you for your question. We carefully examined the absorption spectroscopy of $\text{Fe}_n(\text{bim})_{2n}$ flakes again. The absorption of $\text{Fe}_n(\text{bim})_{2n}$ flakes is indeed very weak throughout the entire spectrum, only with slightly obvious absorption in the ultraviolet range (≤ 390 nm). In the visible range (390-780 nm), the absorption of $\text{Fe}_n(\text{bim})_{2n}$ flakes is less than 2×10^{-2} , which is considerably weak so that interference from background error becomes prominent. To further confirm it, UV-vis spectrum of bulk $\text{Fe}_n(\text{bim})_{2n}$ was recorded as shown below (Fig. R1b), in which its absorption at 600-800 nm is zero. This result clearly suggests that the non-zero absorption at 600-800 nm for the $\text{Fe}_n(\text{bim})_{2n}$ flakes can be attributed to the background error. The background error may arise from the difference between the reference substrate and the test substrate used in the measurement. Tauc plot was further used to determine the optical bandgap of $\text{Fe}_n(\text{bim})_{2n}$ flakes, with extrapolation of the linear region (red line, Fig. R2) estimating of approximately 3.30 eV.

Figure R1. UV-Visible absorption spectra of $\text{Fe}_n(\text{bim})_{2n}$ flakes (a) and bulk $\text{Fe}_n(\text{bim})_{2n}$ (b).

Figure R2. Optical bandgap of $\text{Fe}_n(\text{bim})_{2n}$ flakes.

Reviewer #3 (Remarks to the Author):

(1) Some new data graphics may be updated into text graphics, such as GIWAXS.

ANS: Thank you for your advice. We have updated the GIWAXS graphic into Supporting information as Supplementary Fig. 11.

Supplementary Figure 11. The grazing incidence wide angle X-ray scattering (GIWAXS) image of $\text{Fe}_n(\text{bim})_{2n}$ nanosheets. The GIWAXS characterization was carried out using XEUSS 2.0, but no signals have been detected on the $\text{Fe}_n(\text{bim})_{2n}$ flakes. This is because the brilliance of the radiation is not high enough to detect the diffraction of 2D crystals.

Revisions made:

We have added Supplementary Fig. 11 in supporting information.

(2) Theoretical calculations related to in-plane extension, including DFT, need to be described in detail.

ANS: Thank you for your advice. The in-plane extension growth of $\text{Fe}_n(\text{bim})_{2n}$ was analyzed theoretically by referring to and learning from previously reported works, including ACS Nano 17, 127 (2023), Chem. Rev. 121, 6321 (2021) and J. Am. Chem. Soc. 133, 5009 (2011)), without conducting theoretical calculations. Following your suggestion, a clearer description on in-plane extension growth has been added.

“In addition, similar to most 2D materials, the atoms in a $\text{Fe}_n(\text{bim})_{2n}$ layer interact through strong chemical bonds, while the interactions between layers are weak van der Waals force¹⁰. Previous density functional theory (DFT) studies have shown that the layer–layer interactions in most van der Waals materials are around 0.03-0.2 eV per atom, while the strength of typical chemical bonds is between 2 and 8 eV per atom (Chem. Rev. 121, 6321 (2021)). The strength of intralayer interaction is 1-2 orders of magnitude higher than that of interlayer interaction. Because of the stronger intralayer binding, attaching atoms to an edge of a $\text{Fe}_n(\text{bim})_{2n}$ layer to facilitate the

in-plane extension is much easier than adsorbing atoms on the surface of a $\text{Fe}_n(\text{bim})_{2n}$ layer for the growth of multilayers. More specifically, the nucleation energy for a new layer is the maximum of $\alpha\sqrt{A} - \beta A$, where A is the nucleus size, and the first and second terms represent the positive and negative contributions from the edge and bulk of the nucleus, respectively (J. Am. Chem. Soc. 133, 5009 (2011)). In comparison, the nucleation energy for the growth of an edge (in-plane extension) is the energy of new dangling bonds when forming a unitcell of $\text{Fe}_n(\text{bim})_{2n}$ on the edge (ACS Nano 17, 127, (2023)). Under a near equilibrium growth condition (low β), the critical nucleation size for a new layer will be much larger than a unitcell of $\text{Fe}_n(\text{bim})_{2n}$, and more dangling bonds involves in the formation of a new layer, which results in a higher nucleation energy for a new layer. Therefore, selective in-plane growth of $\text{Fe}_n(\text{bim})_{2n}$ is favourable in an equilibrium reversible thermodynamic process, as the case in CVD growth of most 2D Van der Waals crystals.”

Revisions made:

We added a clearer description on in-plane extension growth of $\text{Fe}_n(\text{bim})_{2n}$ in the text.

“Previous density functional theory (DFT) studies have shown that ... results in a higher nucleation energy for a new layer.” (line 199-214)

(3) The optical microscope image of the sensor device needs to be given in SI.

ANS: Thank you for your reminder. We added the optical microscope image of the sensor device in Supplementary Fig. 18.

Supplementary Figure 18. The optical image of the $\text{Fe}_n(\text{bim})_{2n}/\text{MoS}_2$ sensor. Scale bars, 20 μm .

Revisions made:

We added Supplementary Fig. 18 in supporting information.

Reviewer #4 (Remarks to the Author):

The definition of a single crystal is a crystalline solid for which the periodic and repeated atomic pattern extends throughout its entirety without interruption. (Materials Science and Engineering: An Introduction, 10th Edition, William D. Callister Jr., David G. Rethwisch)

The grain boundaries in Figure 3 a and b are obvious, which means that there are a lot of grains growth on the substrate and the crystal structure is not continuous. Despite those grains with the same crystal orientation, they cannot be called single crystals. Therefore, the optical image of Supplementary Figure 1b may not be evident to prove the grain size is 62 μm .

ANS: Thank you very much for your question. Due to their atomic thicknesses, the $\text{Fe}_n(\text{bim})_{2n}$ flakes transferred onto TEM grid are almost transparent. When TEM images the $\text{Fe}_n(\text{bim})_{2n}$ flakes on TEM grid, it is very easy to confuse the $\text{Fe}_n(\text{bim})_{2n}$ flakes with the TEM grid below. To more clearly observe the $\text{Fe}_n(\text{bim})_{2n}$ flakes, bare TEM grids without loading $\text{Fe}_n(\text{bim})_{2n}$ flakes were imaged by TEM, as shown below. Comparison shows that the boundary-like areas marked with blue arrows in Figs. 3a, b actually belong to the TEM grid below. The area enclosed by a white dashed circle (Fig. R3a) is a crack and fold caused by transfer process, which is common in TEM images of atomically 2D material. There are no grain boundaries observed in the $\text{Fe}_n(\text{bim})_{2n}$ flakes (Figs. 3a, b). Therefore, the crystal structure is continuous in the $\text{Fe}_n(\text{bim})_{2n}$ flakes. According to the definition of a single crystal, the shown $\text{Fe}_n(\text{bim})_{2n}$ flakes with continuous crystal structure and same lattice orientation are single crystals.

Figure R3. TEM images of Figs. 3a (a) and 3b (b), and TEM images of their corresponding bare TEM grid (c, d).

REVIEWER COMMENTS

Reviewer #1 (Remarks to the Author):

I am happy with most of the revisions made by the authors. I would like to thank them for taking my suggestions to improve them. I would like to support its publication now with one last comment on the absorption:

They claimed that "In the visible range (390-780 nm), the absorption of $\text{Fen}(\text{bim})_2\text{n}$ flakes is less than 2×10^{-2} , which is considerably weak so that interference from background error becomes prominent. To further confirm it, UV-vis spectrum of bulk $\text{Fen}(\text{bim})_2\text{n}$ was recorded as shown below (Fig. R1b), in which its absorption at 600-800 nm is zero. This result suggests that the non-zero absorption at 600-800 nm for the $\text{Fen}(\text{bim})_2\text{n}$ flakes can be attributed to the background error" I am afraid that I got the opposite conclusion from the authors: if you get 0 for bulk sample, that means you can substrate the background nicely. This means the absorption in the 600-700 nm is not background. Again, given the average small signal in the entire regime, I think one can not conclude that the bandgap is large based on the current data. Can you do your best to make the background signal free by e.g. measuring in an integrating sphere?

RESPONSE TO REVIEWERS' COMMENTS

Reviewer #1 (Remarks to the Author):

(1) I am happy with most of the revisions made by the authors. I would like to thank them for taking my suggestions to improve them. I would like to support its publication now with one last comment on the absorption:

They claimed that "In the visible range (390-780 nm), the absorption of $\text{Fe}_n(\text{bim})_{2n}$ flakes is less than 2×10^{-2} , which is considerably weak so that interference from background error becomes prominent. To further confirm it, UV-vis spectrum of bulk $\text{Fe}_n(\text{bim})_{2n}$ was recorded as shown below (Fig. R1b), in which its absorption at 600-800 nm is zero. This result suggests that the non-zero absorption at 600-800 nm for the $\text{Fe}_n(\text{bim})_{2n}$ flakes can be attributed to the background error" I am afraid that I got the opposite conclusion from the authors: if you get 0 for bulk sample, that means you can substrate the background nicely. This means the absorption in the 600-700 nm is not background. Again, given the average small signal in the entire regime, I think one can not conclude that the bandgap is large based on the current data. Can you do your best to make the background signal free by e.g. measuring in an integrating sphere?

ANS: Thank you very much for your time and professional advice. We have measured the absorption of $\text{Fe}_n(\text{bim})_{2n}$ flakes with an integrating sphere, as shown below.

Figure R1. UV-visible absorption spectrum of $\text{Fe}_n(\text{bim})_{2n}$ flakes measured with an integrating sphere.

REVIEWERS' COMMENTS

Reviewer #1 (Remarks to the Author):

With the updated absorption spectrum, I think it is clear that the bandgap seems indeed relatively large. I am happy to support its publication in Nature communications in its current form.

RESPONSE TO REVIEWERS' COMMENTS

Reviewer #1 (Remarks to the Author):

With the updated absorption spectrum, I think it is clear that the bandgap seems indeed relatively large. I am happy to support its publication in Nature communications in its current form.

ANS: We thank the reviewer for the professional advice on improving our manuscript.